# Expanding Roles of the E2F-RB-p53 Pathway in Tumor Suppression

**DOI:** 10.3390/biology12121511

**Published:** 2023-12-11

**Authors:** Yaxuan Zhou, Rinka Nakajima, Mashiro Shirasawa, Mariana Fikriyanti, Lin Zhao, Ritsuko Iwanaga, Andrew P. Bradford, Kenta Kurayoshi, Keigo Araki, Kiyoshi Ohtani

**Affiliations:** 1Department of Biomedical Sciences, School of Biological and Environmental Sciences, Kwansei Gakuin University, 1 Gakuen Uegahara, Sanda, Hyogo 669-1330, Japan; hrr53817@kwansei.ac.jp (Y.Z.); gnr59822@kwansei.ac.jp (R.N.); idl05439@kwansei.ac.jp (M.S.); hsj19688@kwansei.ac.jp (M.F.); ght57978@kwansei.ac.jp (L.Z.); 2Department of Obstetrics and Gynecology, University of Colorado School of Medicine, Anschutz Medical Campus, 12800 East 19th Avenue, Aurora, CO 80045, USA; ritsuko.iwanaga@cuanschutz.edu (R.I.); andy.bradford@ucdenver.edu (A.P.B.); 3Division of Molecular Genetics, Cancer Research Institute, Kanazawa University, Kakuma-machi, Kanazawa 920-1192, Japan; kentakurayoshi@staff.kanazawa-u.ac.jp; 4Department of Morphological Biology, Ohu University School of Dentistry, 31-1 Misumido Tomitamachi, Koriyama, Fukushima 963-8611, Japan; keigoaraki.res@gmail.com

**Keywords:** RB, E2F, ARF, MDM2, p53

## Abstract

**Simple Summary:**

The RB and p53 pathways are two major tumor suppressive pathways. Hence, in almost all cancers, both pathways are disabled. E2F transcription factor is under the control of the RB pathway and fulfills pivotal roles in tumor suppression by triggering the p53 pathway upon loss of RB function. Although this notion has been well established, emerging evidence indicates that each player of the E2F-RB-p53 pathway possesses novel functions and regulatory mechanisms. This review aims to introduce the expanding roles of the E2F-RB-p53 pathway in tumor suppression.

**Abstract:**

The transcription factor E2F links the RB pathway to the p53 pathway upon loss of function of pRB, thereby playing a pivotal role in the suppression of tumorigenesis. E2F fulfills a major role in cell proliferation by controlling a variety of growth-associated genes. The activity of E2F is controlled by the tumor suppressor pRB, which binds to E2F and actively suppresses target gene expression, thereby restraining cell proliferation. Signaling pathways originating from growth stimulative and growth suppressive signals converge on pRB (the RB pathway) to regulate E2F activity. In most cancers, the function of pRB is compromised by oncogenic mutations, and E2F activity is enhanced, thereby facilitating cell proliferation to promote tumorigenesis. Upon such events, E2F activates the *Arf* tumor suppressor gene, leading to activation of the tumor suppressor p53 to protect cells from tumorigenesis. ARF inactivates MDM2, which facilitates degradation of p53 through proteasome by ubiquitination (the p53 pathway). P53 suppresses tumorigenesis by inducing cellular senescence or apoptosis. Hence, in almost all cancers, the p53 pathway is also disabled. Here we will introduce the canonical functions of the RB-E2F-p53 pathway first and then the non-classical functions of each component, which may be relevant to cancer biology.

## 1. Introduction

Cancer is a genetic disease caused by mutations, which activate proto-oncogenes and inactivate tumor suppressor genes, leading to unrestrained proliferation of cells. Tumor suppressor genes fulfill crucial roles in the suppression of tumorigenesis by counteracting activation of proto-oncogenes, thereby suppressing cell proliferation. Cells possess two major tumor suppressive pathways to suppress tumorigenesis [1]; one is the retinoblastoma (RB) pathway, and the other is the p53 pathway. The RB pathway restrains cell proliferation through inhibition of the E2 transcription factor (E2F), an activator of growth-related genes, to suppress tumorigenesis. In the resting state of normal cells, RB binds to and inactivates E2F. Growth stimulatory signals and growth suppressive signals converge on RB (the RB pathway), regulating E2F to modulate cell proliferation. Hence, in most cancers, loss of function of the RB pathway, caused by oncogenic changes, facilitates cell proliferation by activating E2F, thereby promoting tumorigenesis [1]. Conversely, upon such events, E2F, liberated from control by RB, activates the *alternative reading frame* (*Arf*) tumor suppressor gene, leading to activation of the tumor suppressor p53 to protect cells from tumorigenesis [2]. Expression of the *Tp53* gene, which codes for p53, is generally constitutive, and expression of p53 is mainly regulated at the protein level. In unstressed cells, p53 is ubiquitinated by mouse double minute 2 (MDM2; in humans, HDM2) and degraded by the proteasome; thereby, the p53 protein level is kept low. ARF sequesters MDM2 to the nucleolus and stabilizes p53 to enhance its activity (the p53 pathway). P53 is a transcription factor and activates genes involved in the induction of cell cycle arrest, resulting in cellular senescence (irreversible cell cycle arrest). P53 also activates genes involved in apoptosis (programmed cell death), resulting in cancer cell death, depending on cellular circumstances. Both cellular senescence and apoptosis contribute to the suppression of tumorigenesis mediated by p53. Hence, in almost all cancers, the p53 pathway is also disabled so that cancer cells can survive [1]. As such, the main role of the RB pathway in the suppression of tumorigenesis is restraining E2F activity to avoid uncontrolled proliferation of cells. The main role of the p53 pathway is inducing cellular senescence or apoptosis upon dysfunction of the RB pathway. The main role of E2F is activation of the p53 pathway upon dysfunction of pRB by oncogenic changes. The roles of these two major tumor suppressive pathways, and those of E2F in linking the two, have been well established. However, recent studies have identified additional functions and regulatory mechanisms of each component of the pathways, such as p53-independent roles of ARF and transcription-independent roles of p53 in the suppression of tumorigenesis as well as the unique properties of E2F in connecting the RB pathway to the p53 pathway. Here, we will summarize the classical roles of the RB-E2F-p53 pathway in tumor suppression and introduce novel functions of each component, focusing on the crucial role of E2F in linking these two major tumor suppressive pathways.

## 2. Classical Views of the RB-E2F-p53 Pathway

### 2.1. The E2F Family of Transcription Factors

Analysis of the molecular mechanism of induction of adenovirus *E2* gene expression by adenovirus E1a identified a cellular transcription factor termed E2 transcription factor (E2F) [3]. Later studies showed that E2F activates a variety of growth-associated genes, thereby playing a central role in cell proliferation [4,5]. In addition, E2F also plays crucial roles in other important biological processes such as apoptosis, tumor suppression, development, differentiation, cellular metabolism, angiogenesis, metastasis, stemness, and others [6,7,8,9,10,11,12,13,14,15,16,17,18,19,20]. E2F is composed of eight family members (E2F1–E2F8), which regulate these biological processes. According to their primary function, E2F1–E2F3a are regarded as transcriptional activators and E2F3b–E2F8 are regarded as transcriptional repressors (Figure 1). Expression of E2F1–E2F3a is induced at the G1/S boundary of the cell cycle by E2F itself [21,22,23]. At the cell cycle phase, retinoblastoma (RB) family proteins (pRB, p107, p130) are inactivated by phosphorylation through cyclin-dependent kinases (CDKs), enabling E2F1–E2F3a to activate target genes. In contrast, E2F3b–E2F5 are expressed throughout the cell cycle and, together with RB family proteins (E2F3b/pRB, E2F4, 5/p130), repress target gene expression in the resting state [24,25,26,27]. In addition, E2F6–E2F8 repress expression of target genes independently of RB family proteins [28,29,30,31]. All E2F family members contain one or two highly conserved DNA-binding domains (DBDs) to bind to their target genes [11,12,16]. E2F1–E2F6 possess a DBD N-terminal to a dimerization domain composed of a leucine zipper (LZ) and a marked box (MB) [10,11,12,16]. E2F1–E2F3 possess a nuclear localization signal (NLS) in the N-terminal region to translocate into the nucleus after synthesis in the cytoplasm to regulate target genes [32,33]. They also have a cyclin A binding site, through which cyclin A/CDK2 binds to and phosphorylates E2F/DP to inhibit DNA-binding activity and terminate E2F activity, resulting in exit from the S phase [34,35]. E2F4 and E2F5 possess bipartite nuclear export signals (NESs) that facilitate their nuclear export after inactivation of p130 [36,37]. E2F1–E2F6 form heterodimers with dimerization partner 1 (DP1) or DP2 through the dimerization domain to bind to target genes with high affinity [38]. E2F7 and E2F8 have two DBDs and bind to target genes independently of DP [29,30,31,39,40]. E2F3c and E2F3d are newly identified isoforms of E2F3a, which do not act as transcription factors [41].

### 2.2. Roles of E2F in Cell Proliferation

E2F activates a variety of growth-associated genes, thereby playing a central role in cell proliferation [4]. Upon growth stimulation, cell cycle progression is promoted under the control of cyclins and CDKs. The progression of the cell cycle through the G1 phase up to the restriction point in the late G1 phase depends on the growth stimulation. Beyond this restriction point, cell cycle progression is independent of growth stimulation and is programmed to proceed autonomously to the end of the M phase. Thus, the restriction point is the critical determinant of whether or not a cell proliferates. E2F and pRB play a central role in controlling this restriction point. E2F induces the expression of growth-associated genes, which are required for passing through the restriction point, such as *cyclin E* and *E2F1*–*E2F3a*, while pRB controls the activity of E2F (see below). E2F also activates genes involved in cell cycle progression throughout the cell cycle, such as those promoting cell cycle progression, DNA synthesis, DNA replication, checkpoints, G2/M phase progression, and others, thereby playing a fundamental role in cell proliferation [4,5].

### 2.3. Regulation of E2F Activity by the RB Pathway

Activator E2Fs are induced at the late G1 phase of the cell cycle, and their activity is controlled by the binding of pRB. The retinoblastoma gene (*Rb1*), which codes for pRB, is the first tumor suppressor gene identified [42,43]. The ability of E2F proteins to activate or repress target gene expression is controlled by interactions with the retinoblastoma (RB) protein family (pRB, p107, and p130) [44]. E2F plays central roles in progression through the restriction point by activating growth-related genes such as *cyclin E* and *E2F1*–*E2F3a*. While in the resting state of the cell cycle (G0 phase), expression of growth-related E2F target genes is suppressed by repressor E2Fs bound by RB family proteins. For example, pRB binds to E2F3b [24], while E2F4 and E2F5 bind to p130, forming the DREAM (DP, Rb-like, E2F and MuvB) complex, which is the major repressive complex in the resting state [5,45,46,47,48]. Binding of the DREAM complex requires the cell cycle genes homology region (CHR) (TTTGAA) or CHR-like elements (CLE) (CTTGAC) adjacent to E2F binding sites, which are recognized by LIN54 of the MuvB complex (LIN52, LIN9, LIN37, LIN54, RBBP4) [46] (Figure 2). pRB and the DREAM complex cooperatively repress cell cycle genes in the resting state [49]. Activity of RB family proteins to bind to and inhibit E2F is regulated by phosphorylation through CDKs [50,51]. CDKs regulate cell cycle progression by transiently binding to their partners, cyclins. Upon growth stimulation of cells, expression of cyclin D is induced, activating CDK4 and CDK6, which phosphorylate and inactivate pRB and p130 [5]. When pRB and p130 are inactivated, repression of E2F target genes by pRB and the DREAM complex is released, and the expression of genes required for S-phase progression and DNA replication, such as *cyclin E*, activator E2Fs (*E2F1*–*E2F3a*), and *CDC6*, is induced [21,22,23,52,53,54,55]. Cyclin E activates CDK2, which also phosphorylates and inactivates pRB, further activating E2F. Activator E2Fs (E2F1–E2F3a) replace repressor E2Fs (E2F4 and E2F5), creating a positive feedback loop promoting the inactivation of pRB and activation of E2F. Cyclin E/CDK2 also phosphorylates p27^Kip1^ (one of the CDK inhibitors) to promote its degradation, creating a positive feedback loop regulating the activation of CDK2 to enable progression into the S phase [55,56]. These positive feedback loops shift the inactivation of pRB and consequent activation of E2F from the D-type CDK, which depends on growth stimulation, to the E-type CDK, which depends on E2F but not on growth stimulation, thereby mediating passage through the critical restriction point [57,58,59].

Interestingly, the *Rb1* gene is a target of E2F [60,61,62], and pRB expression is increased around the G1/S boundary of the cell cycle [63]. Moreover, the amount of pRB/E2F complexes also increases at the G1/S boundary, as shown by gel mobility shift assays using a typical E2F binding site [64]. This observation suggests that not all pRB molecules are hyper-phosphorylated and inactivated around the G1/S boundary, with a significant fraction of pRB remaining hypo-phosphorylated and active, thereby modulating E2F activity according to the relative magnitude of growth stimulation and activity of CDKs.

Activator E2Fs activate *E2F7* and *E2F8* genes, and the resulting accumulation of E2F7 and E2F8 suppresses the expression of a subset of E2F target genes, including activator E2Fs, forming a negative feedback loop in the control of E2F activity [16,28,29,30,31]. After cells have entered the S phase, cyclin E is degraded, and CDK2 then associates with cyclin A. Cyclin A is another cyclin required for DNA replication, whose synthesis is initiated during late G1 and whose associated kinase activity is first detected in the S phase. As the S phase progresses to the end of the S phase, E2F activity declines due to F box protein cyclin F (CCNF)-mediated recognition and ubiquitination of activator E2Fs (E2F1–E2F3a) by Skp, Cullin, F-box (SCF) ubiquitin ligase for degradation by proteasome, in addition to inhibition of the expression of activator E2F by accumulated E2F7 and E2F8 [16,40,65,66,67]. Furthermore, activator E2Fs are phosphorylated by cyclin A/CDK2 bound to a cyclin A binding site, which decreases the DNA-binding activity of activator E2Fs [34,35]. Once the cells enter into the G2 phase, E2F4 undergoes nuclear localization and suppresses E2F targets in the late G2 phase [16]. Then, CDC2 associates with cyclin B to promote entry into mitosis [68]. Meanwhile, E2F7 and E2F8 are degraded by the ubiquitin proteasome system to reestablish the E2F transcriptional drive, promoting reentry into the next cell cycle [66,69].
Figure 2Regulation of E2F by RB family members. In the resting state of the cell cycle, RB binds to E2F to suppress the transcription of target genes. pRB binds to E2F3b, and p130 binds to E2F4 and E2F5. P130 bound to E2F4 and E2F5 interacts with the MuvB complex to form the DREAM complex, which constitutes the major repressor of E2F target genes in quiescent cells. Phosphorylation of pRB and p130 by growth stimulation inactivates pRB and p130 to relieve E2Fs from their repression, thereby activating E2Fs. Induced activator type E2F1–E2F3a replace repressor E2Fs, and cyclin E activates CDK2, which further phosphorylates and inactivates RB. This forms a positive feedback loop in the inactivation of RB and activation of E2F, controlling passage through the cell cycle restriction point.
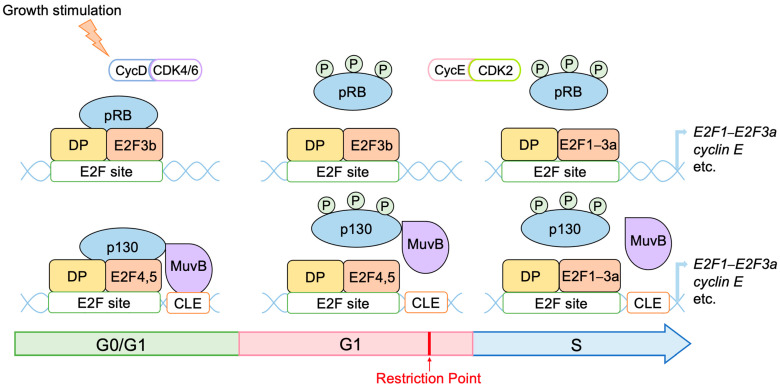


### 2.4. Defects in the RB Pathway in Cancer Cells

Due to the crucial roles played by the RB pathway in suppressing E2F activity and consequently cell proliferation, it is well understood that oncogenic mutations that disrupt the function of the pathway are frequently found [70]. In most cancers, the RB pathway is disabled by oncogenic mutations, and the E2F activity is enhanced [1,56,70]. The tumor suppressor gene *Rb1* is the causative gene of familial retinoblastoma and is the first representative tumor suppressor gene identified [71]. About 30% of cancers have mutations in the *Rb1* gene, which promotes cell proliferation and tumorigenesis by activating E2F [1]. Other than deletion or mutation of the *Rb1* gene, the most frequent mutation locus in human cancers that disables the RB pathway is the *CDKN2A* locus coding for the p16^INK4a^ CDK inhibitor (CKI) [72]. The p16^INK4a^ controls cyclin D/CDK4, 6 kinase activity. Thus, loss of p16^INK4a^ function enhances cyclin D/CDK4 activity, leading to RB inactivation and E2F activation. Finally, over-expression of the D-type cyclins, as well as mutation of CDK4, which makes CDK4 insensitive to CKIs, also increases cyclin D/CDK4, 6 activity and disables the pathway [72,73].

Proto-oncogenes code for factors that play major roles in transmitting growth signals to the nucleus, such as growth factors, signal transducers, and nuclear transcription factors [74]. Growth signals converge on the expression of cyclin D, which functions as a sensor of growth signals and leads to inactivation of RB [72]. Mammalian genomes contain a range of proto-oncogenes that control normal cell proliferation [74]. There are two types of changes that lead to the activation of proto-oncogenes: one is mutation of the coding region of the genes, which leads to structure and functional changes of the encoded proteins, such as Ras, and the other is gene amplification or chromosomal translocation that leads to over-expression of the proteins, such as c-Myc [75,76]. In this way, activation of proto-oncogenes as well as inactivation of tumor suppressor genes, by environmental factors or genetic mutations, lead to tumorigenesis [70].

### 2.5. E2F Links the RB Pathway to the p53 Pathway

In addition to its important functions in cell proliferation, E2F can also induce apoptosis or cellular senescence, which play crucial roles in tumor suppression [8,9,11,77]. In most cancers, the RB pathway is defective due to oncogenic changes such as deletion or mutation of pRB, over-expression of cyclin D, mutation of CDK4 making it insensitive to CKIs, and deletion of CKIs [1,56,70,72]. Therefore, activated E2F induces the expression of growth-associated genes, promoting cell proliferation and contributing to tumorigenesis. However, in addition to growth-associated genes, E2F activated by out of control by pRB (deregulated E2F) also activates the *alternative reading frame* (*Arf*) gene, an upstream activator of the tumor suppressor p53 [2], whose loss contributes significantly to tumorigenesis [78] (Figure 3). Therefore, E2F plays a pivotal role in the suppression of tumorigenesis by linking the RB pathway to the p53 pathway, two major regulators/effectors of cell growth and survival.

### 2.6. The p53 Pathway

The tumor suppressor p53, together with pRB, plays a central role in tumor suppressions. Both proteins are also key regulators of cell proliferation [1]. The activity of pRB and p53 is regulated by two products of the *CDKN2A* (*INK4a*/*ARF*) locus, p16^INK4a^ and p14^ARF^ (p19^ARF^ in mice), respectively. Hence, p16^INK4a^ and p14^ARF^ play crucial roles in tumor surveillance [79,80,81] (Figure 4). P53 is now regarded as a factor that protects cells from a variety of cellular stresses, amongst which, oncogenic stress and DNA damage have been extensively characterized. When the RB pathway is rendered defective by oncogenic changes, E2F is activated out of control by RB and induces expression of ARF. ARF in turn inhibits mouse double minute 2 (MDM2) (HDM2 in humans), a negative regulator of p53, resulting in the activation of p53.

Expression of the *Tp53* gene, which codes for p53, is relatively constitutive, and p53 expression is mainly regulated at the protein level. In unstressed cells, p53 is expressed at very low levels with a short half-life (minutes). This is because p53 is continuously ubiquitinated and degraded through the proteasome. The principal E3 ubiquitin ligase for p53 is the oncoprotein MDM2 [82]. MDM2 negatively regulates p53 in multiple ways, such as interfering with the transactivation of target genes by binding to p53 [83,84], the degradation of p53 via its intrinsic ubiquitin ligase activity [85], and the translocation of p53 from the nucleus to the cytoplasm to facilitate proteasome-mediated degradation [86,87] (Figure 5).

The extreme N-terminal region of ARF, encoded by exon 1, contributes strongly to MDM2 binding and is most conserved between species. ARF interaction sequesters MDM2 into the nucleolus, thereby suppressing ubiquitination of p53 by MDM2 in the nucleoplasm, stabilizing p53 protein [88,89,90,91,92,93,94,95]. However, ARF mutants, which lack the ability to sequester p53 into the nucleolus, are still able to stabilize p53 [96], suggesting that nucleolar sequestration of MDM2 by ARF facilitates but is not essential for p53 stabilization. The binding of ARF also inhibits the intrinsic ubiquitin-ligase activity of MDM2 [97,98].

To ensure appropriate control and coordination of p53 activity under normal cell conditions and in response to a variety of cellular stresses, the activation of p53 triggers feedback mechanisms to negatively regulate its activity. The activation of p53 enhances the expression of its negative regulator MDM2 [99] and suppresses the expression of its positive regulator ARF [100], thereby fine-tuning its own activity (Figure 5).

In addition to oncogenic stresses, p53 is activated when DNA is damaged by extra- and intra-cellular genotoxic stresses such as UV irradiation, carcinogens, and reactive oxygen species (ROS) produced during respiratory metabolism. DNA damage also results in p53 protein accumulation but through a mechanism distinct from that in response to oncogenic stresses. When DNA is damaged, ataxia telangiectasia mutated (ATM) and ataxia telangiectasia and Rad3-related (ATR) kinases are activated to phosphorylate and activate checkpoint kinase 1 (CHK1) and CHK2. These kinases phosphorylate p53 at Ser15 and Ser20, which reduces the interaction of p53 with MDM2 and stabilizes p53 (Figure 6). Phosphorylated p53 undergoes a conformational change, mediated by the prolyl isomerase Pin1, which is required for efficient escape from MDM2-mediated degradation and transcriptional activation [101,102]. Accumulated p53 functions as a transcription factor and activates target genes related to G1 arrest, such as *p21^Cip1^* [103,104], arresting the cell cycle at the G1 phase in order to facilitate the repair of damaged DNA. In the event that DNA damage is extensive and cannot be repaired, to avoid generation of cells with mutated DNA, p53 is additionally phosphorylated on Ser46, resulting in the activation of pro-apoptotic target genes, such as *Bax* [105], causing cell death. Thus, p53 plays a crucial role in maintaining genome stability by mediating cell-cycle arrest or apoptosis in response to genotoxic stresses and is hence dubbed “the guardian of the genome” [106].

In addition to MDM2, MDM4 also binds to p53 and inhibits its transcriptional activity. However, MDM4 does not ubiquitinate p53 or facilitate its degradation. In spite of this, *Mdm4* knockout mice are embryonic lethal, although at later stages than that of *Mdm2*, and this phenotype is completely abrogated by combined knockout of *Tp53* [107], indicating that MDM4 also plays a significant role in controlling p53 activity. MDM4 forms a heterodimer with MDM2, which is more prevalent than the MDM2 homodimer and has a much higher affinity than the MDM2 homodimer for p53 [108,109]. MDM4 binding enhances the ubiquitin ligase activity of MDM2 by recruiting the ubiquitin-conjugating enzyme UbcH5c through the C-terminal region of MDM4 [109]. Accordingly, knock-in of a mutant MDM4, which cannot bind to MDM2 but retains the ability to bind to p53, is also embryonic lethal in mice, concomitant with elevated levels and activity of p53 [110]. These results indicate that MDM4, in addition to inhibiting transcriptional activity of p53, contributes to the degradation of p53 by facilitating the E3 ligase activity of MDM2. FAM193A interacts with and destabilizes MDM4, thereby enhancing p53 transcriptional activity [111].

Since ubiquitination of p53 facilitates degradation, de-ubiquitination of p53 is thought to stabilize p53. Herpesvirus-associated ubiquitin-specific protease (HAUSP) interacts with and stabilizes p53 by deubiquitinating p53 [112]. Intriguingly, HAUSP also stabilizes MDM2 by de-ubiquitinating self-ubiquitinated MDM2 [113,114] and functions in the MDM2-p53-HAUSP trimeric complex [115]. Thus, the effects of HAUSP on p53 stabilization may be context-dependent. For example, an shRNA-mediated reduction in HAUSP levels destabilizes p53, whilst complete knockdown of HAUSP results in the stabilization of p53 due to the concomitant loss of MDM2 [113].

In addition to the phosphorylation and ubiquitination of p53, other modifications such as SUMOylation (conjugation of the small ubiquitin-like modifier (SUMO) protein) and NEDDylation (conjugation of the ubiquitin-like protein NEDD8) also regulate p53 activity (Figure 7). P14^ARF^ promotes SUMOylation of MDM2, and MDM2 together with p14^ARF^ facilitates SUMOylation of p53, dependent on the residues in exon 2 of p14^ARF^ involved in nucleolar localization along with MDM2 and p53 [116,117]. SUMOylation of p53 enhances its transcriptional activity. ARF and MDM2 can also promote the conjugation of SUMO-2/3 to human p53, but not mouse p53, and reduce both the activation and repression of a subset of p53-target genes [118]. Promyelocytic leukemia protein (PML) nuclear bodies (NBs) recruit a variety of proteins and regulate their activity by SUMOylation and thus are involved in a number of cellular functions, including apoptosis and cellular senescence. PML splice variant IV binds ARF and enhances SUMOylation of p53, leading to the stabilization and activation of p53 [119]. ARF interacts with and stabilizes the SUMO-conjugating enzyme UBC9 associated with NB, which may explain PML IV-mediated SUMOylation [119]. MDM2 also promotes NEDDylation of p53 and inhibits its transcriptional activity [120].

Induction of the CDK inhibitor p21^Cip1^ plays a pivotal role in p53-mediated cell cycle arrest in G1 by regulating CDK activities [103,104]. The inhibition of CDK activities by p21^Cip1^ activates the DREAM complex by reducing the phosphorylation of p130, and it suppresses the expression of E2F target genes involved in both G1/S and G2/M cell cycle progression [46,47]. The *phosphatase of regenerating liver-3* (*Prl-3*) and *protein tyrosine phosphatase receptor type V* (*Ptprv*) are also p53-inducible genes, which can induce G1 cell cycle arrest [121,122]. P53 can also cause G2/M cell cycle arrest by inhibiting cell division cycle 2 (CDC2) activity, which is required for the G2/M transition [123]. P53 increases the expression of 14-3-3 σ, which suppresses CDC2 activity by inhibiting the formation of the CDC2-cyclin B1 complex [124,125]. P53-induced growth arrest and DNA damage inducible 45 (GADD45) and Reprimo can also induce G2/M arrest by inhibiting CDC2 activity [126,127].

The mechanistic target of rapamycin (mTOR) protein kinase plays a crucial role in nutrient and growth factor signaling. For the activation of mTORC1, one of the two mTOR complexes (mTORC) that is sensitive to rapamycin, guanosine triphosphate (GTP)-bound Rheb is required (Figure 8). Tuberous sclerosis complex 1 (TSC1) and TSC2 form heterodimers and inactivate Rheb by stimulating conversion to the guanosine diphosphate (GDP)-bound form by GTPase-activating protein (GAP) activity. P53 suppresses the mTOR pathway by transcriptionally inducing the expression of Sestrin1 and Sestrin2, which activate the adenosine monophosphate (AMP)-responsive protein kinase (AMPK). AMPK phosphorylates TSC2 and stimulates its GAP activity, thereby suppressing Rheb [128,129]. Phosphatase and Tensin homolog deleted from chromosome 10 (PTEN) tumor suppressor is a negative regulator of phosphatidylinositol-3 kinase (PI3K)/AKT/protein kinase B (PKB)-mediated cell survival and growth-promoting signaling pathway. P53 transcriptionally induces the expression of PTEN, which is required for p53-mediated apoptosis in immortalized mouse embryonic fibroblasts [130].

In addition to inducing growth arrest, p53 contributes to tumor suppression by inducing apoptosis. The phosphorylation of Ser46 of p53 changes its target genes from those involved in cell cycle arrest to those involved in apoptosis [131,132,133]. Thus, the kinases responsible for the phosphorylation of Ser46 play important roles in the commitment to cell death. Several kinases have been implicated in p53 Ser46 phosphorylation. For example, after UV irradiation, homeodomain-interacting protein kinase-2 (HIPK2) binds to and phosphorylates p53 at Ser46, facilitating the activation of p53-dependent transcription and apoptotic pathways [134]. In addition, upon genotoxic stress, p53 induces the expression of p53-dependent damage-inducible nuclear protein 1 (p53DINP1), which facilitates the phosphorylation of p53 at Ser46, the induction of p53-regulated apoptosis-inducing protein 1 (p53AIP1), and apoptosis. The P53DINP1-containing complex phosphorylates p53 at Ser46, suggesting that p53DINP1 functions as a cofactor for the putative kinase, which phosphorylates p53 at Ser46 [135]. Protein kinase C δ (PKCδ) associates with p53DINP1 to facilitate the phosphorylation of p53 at Ser46 and apoptosis [136]. Lastly, in response to genotoxic stress, the dual-specificity tyrosine-phosphorylation-regulated kinase 2 (DYRK2) translocates into the nucleus and directly phosphorylates p53 at Ser46, thereby inducing p53AIP1 expression and apoptosis [137].

The induction of apoptosis by p53 is mainly mediated through the mitochondria [138,139]. The release of cytochrome *c* from the inter membranous space of the mitochondria into the cytosol triggers a cascade of apoptosis-inducing processes, where cytochrome *c* complexes with apoptotic peptidase activating factor 1 (APAF-1) to form the apoptosome, which activates caspase-9 to initiate the cascade (Figure 9). Thus, mitochondrial outer membrane permeability (MOMP) is the critical determinant of whether to induce apoptosis or not. The outer mitochondrial membrane is stabilized by anti-apoptotic B-cell lymphoma 2 (BCL-2) family members, such as BCL-2, B-cell lymphoma-extra large (BCL-xL), and myeloid cell leukemia 1 (MCL-1), and it is destabilized by pro-apoptotic BCL-2 family members, which are grouped into pore-forming proteins and BCL-2 homology domain 3 (BH3)-only proteins. Pore-forming proteins consist of BCL-2-associated X (BAX) and BCL-2 homologous antagonist/killer (BAK). BCL-2 homology domain 3 (BH3)-only proteins comprise BCL-2 agonist of cell death (BAD), BCL-2 interacting mediator of cell death (BIM), NOXA, and p53 up-regulated modulator of apoptosis (PUMA). BH3-only proteins bind to and activate BAX and BAK to form a pore complex that results in release of cytochrome *c* into the cytoplasm. Conversely, anti-apoptotic BCL-2 family members bind to BH3-only proteins and sequester them from BAX and BAK, thereby stabilizing the outer mitochondrial membrane. P53 transcriptionally induces the expression of many of the pro-apoptotic BCL-2 family members, such as BAX, NOXA, PUMA, and APAF-1 [105,131,140,141,142,143]. P53 also reduces the expression of anti-apoptotic BCL-2 family members, such as BCL-2, by increasing transcription of microRNAs (miRNAs) such as miR-15b, miR-16, miR-34a, and miR-1915 [144,145,146,147,148,149,150,151]. In addition, Amphiregulin (AREG), induced by p53, interacts with an RNA helicase DEAD-box helicase 5 (DDX5) and stimulates the generation of miR-15a to reduce the expression of anti-apoptotic protein BCL-2 [152]. Taken together, p53 plays a central role in the induction of apoptosis by promoting the release of cytochrome *c* from the mitochondria and the formation of the apoptosome (Figure 9).

Since p53 plays a crucial role in maintaining genome stability, it has long been considered that the primary role of p53, as a tumor suppressor, is to prevent the generation of cells with mutations in proto-oncogenes and/or tumor suppressor genes. However, the elucidation of the ARF-MDM2-p53 pathway, with the *Arf* gene serving as a sensor of oncogenic stresses, raised the question as to which is more important for tumor suppression: DNA damage sensing (ATM/ATR-CHK1/CHK2-p53 pathway) or oncogenic stress responses (ARF-MDM2-p53 pathway). To address this question, a mouse model was established in which p53 function could be switched on and off in vivo by fusing p53 with estrogen receptor α, which, in the absence of a hormone, sequesters p53 in the cytoplasm until it is activated by the addition of 4-hydroxy-tamoxifen (4-OHT), an estrogen analogue, in feeding water. Using this model, the incidence of lymphoma induced by whole-body irradiation was examined by switching on p53 function at the time of irradiation or at a later time after the acute DNA damage response subsided. Switching on p53 function concurrently with irradiation had no effect on tumor incidence, whereas activating p53 at the later timepoint significantly suppressed tumorigenesis. Moreover, this effect was absolutely dependent on p19^ARF^ [153]. In a similar experimental approach, both p53 alleles were somatically deleted in mice at various ages. Deletion of p53 at 3 months of age showed a longer tumor latency compared to deletion at 6 and 12 months of age. These results suggest that tissues accumulate oncogenic mutations with age, which are monitored by p53. In addition, p53 was deleted in combination with ionizing radiation (IR). The presence of p53 during IR treatment showed no effect on IR-induced tumor latency, supporting that the immediate DNA damage response does not contribute to tumor suppression [154]. In another model, p53 super mice, which carry an additional copy of the *Tp53* gene, were generated. Wild-type and p53 super mice were treated with DNA-damaging agent 3-methyl cholanthrene, which generates mutations and facilitates tumorigenesis. p53 super mice showed greater resistance to the agent than wild-type, whereas the effect of additional p53 was lost in the absence of ARF [155]. Taken together, these observations indicate that oncogenic signaling is critical for triggering tumor suppression by p53, whereas the acute response of p53 to DNA damage has little effect, underscoring the crucial role of the *Arf* gene in tumor suppression by monitoring oncogenic changes and activating p53 (Figure 10).

## 3. Non-Classical Functions of Each Component of the RB-E2F-ARF-MDM2-p53 Pathway

Although the roles of the RB-E2F-ARF-MDM2-p53 pathway in tumor suppression have been well established, accumulating evidence indicates that each component of the pathway has distinct functions, which do not depend on downstream factors. The contribution of these non-classical functions to tumor suppression or promotion has been discerned in mouse models, in which the downstream factor is knocked out. In addition, there are some reports that are contradictory to or challenge the classical views. We will introduce non-classical functions of each component of the RB-E2F-ARF-MDM2-p53 pathway and review the findings that challenge the classical paradigm.

### 3.1. Non-Classical Functions of RB

pRB has 14 sites, which are phosphorylated by D-type CDKs. It has been widely accepted that hypo-phosphorylated pRB can bind to and inhibit E2F, whilst hyper-phosphorylation of pRB abolishes binding to E2F. Cyclin D/CDK4, 6 inactivates pRB during the early G1 phase by progressive multi-site phosphorylation of 14 sites to release E2F transcription factors from RB inhibition. However, recent evidence suggests that cyclin D/CDK4, 6 can also activate pRB to bind to and inhibit E2F by phosphorylation at one of the 14 phosphorylation sites (mono-phosphorylation) during the early G1 phase [156] (Figure 11). Utilizing two-dimensional isoelectric focusing, it has been shown that cells synchronized in the early G1 phase contain exclusively mono-phosphorylated pRB at one of the 14 different phosphorylation sites, catalyzed by cyclin D/CDK4, 6. Moreover, upon DNA damage, mono-phosphorylated pRB, at these distinct phosphorylation sites, that is fully functional in binding to E2F, is observed. At the late G1 restriction point, when the expression of cyclin E and activator E2Fs is induced, cyclin E/Cdk2 inactivates pRB by hyper-phosphorylation, releasing activator E2Fs from pRB binding [156] (Figure 11). These observations change our understanding of the roles of cyclin D/CDK4, 6 in the regulation of pRB activity with respect to E2F.

E2F transcriptionally induces the expression of genes involved in checkpoints and DNA repair. In addition, E2F1 and pRB localize to sites of DNA damage and directly promote DNA repair [157,158,159] (Figure 12). Upon DNA damage, ATM phosphorylates E2F1 at Ser31 [160], and phosphorylated E2F1 binds to a breast cancer susceptibility gene 1 (BRCA1) C terminus (BRCT) domain of DNA topoisomerase 2-binding protein 1 (TopBP1) at the sites of DNA damage [161]. CHK1/CHK2 phosphorylate pRB at Ser612 and facilitate the binding of pRB to E2F1 [162], leading to the co-localization of pRB with E2F1 at sites of DNA damage. E2F1 and pRB promote DNA repair by several mechanisms. pRB recruits chromatin remodeling factors such as brahma (BRM)/SWItch 2 (SWI2)-related gene 1 (BRG1), which facilitates DNA end resection and homologous recombination repair of DNA double strand breaks (DSB) [163]. PRB also recruits C-terminal-binding protein (CtBP)-interacting protein (CtIP) [164], which also facilitates DSB end resection [165]. E2F1 and pRB promote the loading of the meiotic recombination 11 (MRE11)-RAD50-Nijmegen breakage syndrome 1 (NBS1) (MRN) complex to sites of DNA double strand breaks and facilitate DNA end resection and the activation of ATM [166,167]. E2F1 recruits acetyltransferases p300 and cAMP response element binding protein (CREB)-binding protein (CBP), which facilitates DNA repair by acetylating histones at DNA double-strand breaks [167]. E2F1 co-localizes with RAD51 and replication protein A (RPA) at the sites of DNA DSB and facilitates RAD51-mediated DNA repair [166,168]. E2F1 also localizes to sites of UV-induced DNA damage, which is primarily pyrimidine dimers formed in one DNA strand, rather than double strand DNA breaks, and enhances nucleotide excision repair by recruitment of nucleotide excision repair (NER) factors, such as xeroderma pigmentosum, complementation group C (XPC), and XPA [169,170]. These results indicate that E2F1, together with pRB, directly promotes DNA repair, contributing to genome stability (Figure 12).

This review focuses on the role of E2F in linking the RB and p53 pathways; the non-canonical functions of pRB are the subject of other comprehensive reviews [13,14,171,172,173,174,175,176,177,178,179,180].
Figure 12pRB and E2F1 directly contribute to genome stability. E2F1, and consequently pRB, are recruited to the sites of DNA damage through TopBP1, which is phosphorylated by ATM upon DNA damage. E2F1 and pRB contribute to DNA repair by recruiting histone acetyltransferase (CBP/p300) and chromatin remodeling factors such as BRG1, respectively.
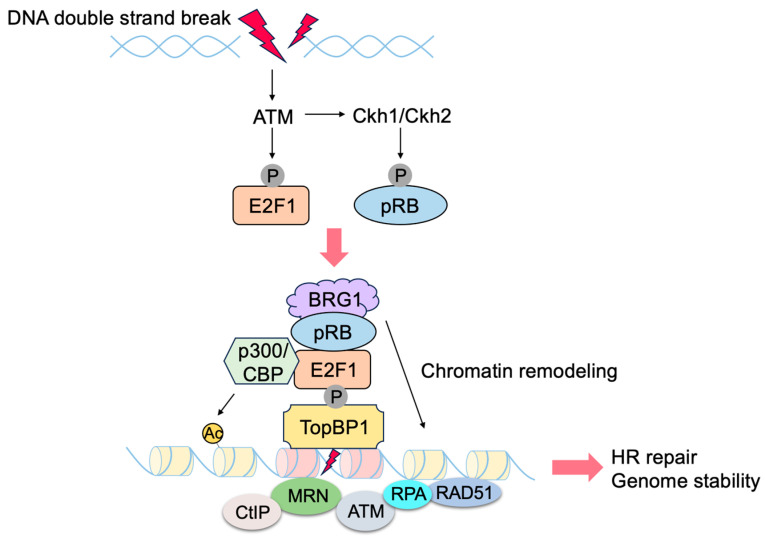


### 3.2. Non-Classical Functions of E2F

#### 3.2.1. Unique Properties of E2F in Linking the RB Pathway to the p53 Pathway

E2F activates a group of growth-related genes upon growth stimulation, thereby playing a central role in cell proliferation. Moreover, upon dysfunction of pRB by oncogenic changes, E2F activates the *Arf* tumor suppressor gene to link the RB pathway to the p53 pathway, two major pathways for tumor suppression, thereby playing a crucial role in the suppression of tumorigenesis against oncogenic changes. Intriguingly, serum stimulation, a physiological growth stimulation of fibroblasts, does not activate the *Arf* gene, whereas the gene is activated by the over-expression of E2F1 or forced inactivation of pRB by adenovirus E1a or knockdown of pRB by shRNA, which mimic the dysfunction of pRB [181] (Figure 13). These results indicate that E2F activates the p53 pathway only when the RB pathway is disabled by oncogenic changes to preserve normal cell proliferation in response to physiologic growth signals. Since growth stimulation does not induce deregulated E2F activity, the *Arf* gene is not activated by E2F in normal growing cells. Together, deregulated E2F activity, which activates the *Arf* gene, is a unique characteristic of cancer cells [181].

The threshold model was suggested to explain the differential regulation of growth-related genes and pro-apoptotic genes by E2F and seems to be widely accepted [10]. It proposes that when the amount of E2F released from RB family members exceeds the first threshold, growth-related genes are activated. Once the amount of free E2F exceeds the second threshold, which is higher than the first one, pro-apoptotic genes are activated in addition to growth-related genes. However, results contradictory to this model have been reported [182]. Activator E2Fs require heterodimeric partner DP to bind to target sequences [38]. According to the threshold model, it is expected that, when the expression of DP is knocked down, the activation of pro-apoptotic genes by over-expressed E2F1 is compromised first, followed by that of growth-related genes. In contrast to this expectation, the knockdown of DP expression significantly reduced E2F1 activation of the growth-related *CDC6* gene but had no effect on the activation of the *Arf* gene. These results indicate that the regulation of the *Arf* gene, by deregulated E2F, cannot be explained by the amount of free E2F. They also indicate that DP is not necessary for E2F to activate the *Arf* gene, building up a distinct activity that activates growth-related genes, which strictly depends on DP [182] (Figure 13). Hence, deregulated E2F activity, which activates pro-apoptotic genes such as *Arf*, is distinct from that which activates growth-related genes.
Figure 13Distinct regulation of the *Arf* gene by E2F. Deregulated E2F activity activates the *Arf* gene, whereas physiological E2F activity induced by growth stimulation does not activate the gene. E2F strictly depends on the heterodimeric partner DP to activate growth-related genes, whereas E2F does not depend on DP to activate the *Arf* gene. Red cross on RB indicates dysfunction of RB and that on dashed line indicates physiologically induced E2F does not activate tumor suppressor genes.
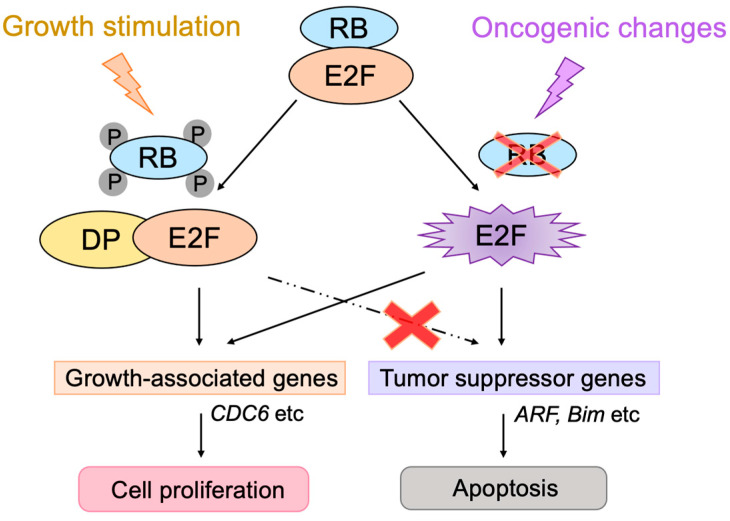


#### 3.2.2. E2F1 Targets Involved in p53-Independent Pathways for the Induction of Apoptosis

ARF can trigger cellular senescence or apoptosis by activating p53. In addition to the *Arf* gene, there are several tumor suppressor genes such as *p27^Kip1^*, *Tap73*, and *Bim*, which are activated by deregulated E2F activity but not by physiological E2F activity [183,184,185]. The induction of p27^Kip1^ expression by deregulated E2F [183] inhibits CDKs, thereby restraining cell cycle progression [177]. TAp73 is a p53 family member, which can activate p53 target genes independent of p53. The *Tap73* gene is an E2F target [186,187] that is specifically activated by deregulated E2F and not by physiological E2F induced by growth stimulation [184]. In addition, there are nine novel targets (*Bim*, *Rassf1*, *Ppp1r13b*, *Jmy*, *Moap1*, *Rbm38*, *Abtb1*, *Rbbp4*, and *Rbbp7*) that are specifically activated by deregulated E2F [185]. Bcl-2 Interacting Mediator of cell death (BIM) is a member of the BH3-only family and destabilizes the mitochondrial outer membrane to release cytochrome *c* and trigger apoptosis. Thus, it is expected that deregulated E2F contributes to the induction of cell cycle arrest or apoptosis, not depending on p53, through alternate pathways mediated by p27^Kip1^, TAp73, BIM, and other products of the genes listed above, which are specifically induced by deregulated E2F1 (Figure 14).

#### 3.2.3. Cancer-Cell-Specific Deregulated E2F Activity as a Cancer-Cell-Specific Targeting Tool

The main obstacles in cancer treatment are the side effects caused by current therapies, which also damage normal cells. To avoid these side effects, we have to damage specifically cancer cells and preserve normal growing cells. Almost without exception, the RB pathway is disabled by oncogenic mutations, and E2F activity is enhanced [1]. Enhanced E2F activity promotes a variety of oncogenic processes such as cell proliferation, thereby contributing to tumorigenesis. Therefore, enhanced E2F activity in cancer cells is a fascinating means to specifically target cancer cells [188,189]. Enhanced E2F activity in cancer cells has been utilized to express cytotoxic genes (suicide gene therapy) or viral genes essential for the replication of viruses (oncolytic virotherapy) by utilizing growth-related E2F targets, such as the E2F1 promoter [190,191,192]. However, E2F activity is also enhanced in normal growing cells, since growth stimulation also activates E2F [21]. Thus, this approach may also adversely affect normal growing cells.

E2F activity that activates growth-related genes is enhanced in both cancer cells and normal growing cells. In contrast, E2F activity that activates the *Arf* gene specifically exists in cancer cells, since growth stimulation does not activate the tumor suppressor gene in normal growing cells [181] (Figure 13). Upon dysfunction of pRB by oncogenic changes, deregulated E2F activates the *Arf* gene and consequently p53 to induce cellular senescence or apoptosis, protecting cells from tumorigenesis [2] (Figure 3). In almost all cancers, the ARF-p53 pathway is also disabled, and cancer cells survive [1], leaving deregulated E2F activity tolerated in cancer cells [181] (Figure 15). Thus, deregulated E2F activity that activates the *Arf* gene in cancer cells seems like a more attractive means to specifically target cancer cells than simply enhanced E2F activity that activates growth-related genes. Indeed, it is reported that the ARF promoter, which is not activated by growth stimulation, exhibits lower activity than the E2F1 promoter, which is activated by growth stimulation, in normal growing cells. However, the ARF promoter showed comparable activity to the E2F1 promoter in cancer cells, thereby showing higher cancer cell specificity than the E2F1 promoter [193]. Similarly, recombinant adenovirus expressing the cytotoxic *HSV-TK* gene under the control of the ARF promoter exhibited lower cytotoxicity than that with the E2F1 promoter in normal growing cells. However, both constructs showed similar cytotoxicity in cancer cell lines [193]. These observations suggest that utilizing deregulated E2F activity by the ARF promoter is superior to that by the E2F1 promoter to drive gene expression specifically in cancer cells [194].

#### 3.2.4. E2F3d, a Novel Member of the E2F3 Family, Mediates Hypoxia-Induced Mitophagy in Cancer Cells

In addition to E2F3a and E2F3b, novel members of the E2F3 family, E2F3c and E2F3d have been identified [41]. Both isoforms are generated by alternative splicing of E2F3a mRNA, and thus their expression parallels that of E2F3a. Intriguingly, both isoforms lack NLS and DBD and localize in the cytoplasm. E2F3d localizes to the outer membrane of the mitochondria and has an LC3-interacting region (LIR) motif created by a shift in the reading frame due to alternative splicing in its cytosolic domain. These observations suggest the involvement of E2F3d in mitochondrial autophagy (mitophagy). Indeed, the over-expression of E2F3d induces mitochondrial fragmentation and mitophagy. Conversely, the knockdown of E2F3s suppresses mitophagy induced by hypoxia and increases intracellular levels of reactive oxygen species in cancer cells. These results indicate that E2F3d facilitates mitophagy upon hypoxia in cancer cells [41] (Figure 16).

This review focuses on the roles of E2F in linking the RB and p53 pathways; other comprehensive reviews have addressed the non-canonical functions of E2F [10,12,14,15,16,17,18,19,176,195].
Figure 16E2F3d facilitates mitophagy upon hypoxia in cancer cells. E2F3d does not possess DBD due to alternative splicing and localizes at the outer membrane of the mitochondria. Due to the frame shift caused by alternative splicing, E2F3d possesses an LC3-interacting region (LIR) motif at the cytosolic domain and functions as a mitophagy receptor upon hypoxia in cancer cells.
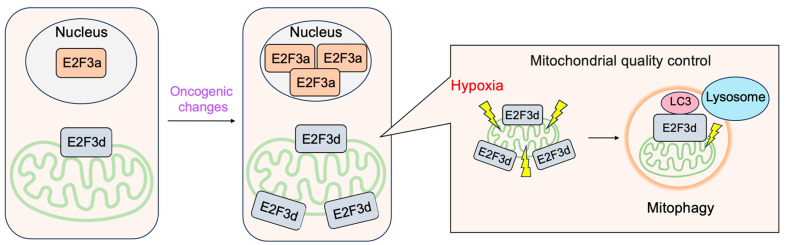


### 3.3. p53-Independent Functions of ARF in Tumor Suppression

Although p53 is a critical element in ARF-mediated tumor suppression, many lines of evidence indicate that ARF has other important tumor suppressor functions, independent of p53 [196,197]. *Arf*, *Tp53*, and *Mdm2* triple knockout mice show a higher frequency of tumor formation than *Tp53* and *Mdm2* double knockout mice, demonstrating that ARF can function independently of MDM2 and p53 in tumor suppression [198]. P14^ARF^ inhibits the proliferation of p53-null human cancer cell lines by inducing G2/M arrest and subsequently apoptosis in tissue cultures and in mouse xenograft models [199]. The introduction of p14^ARF^ into human *Arf*-silenced U-2 OS and p53(−/−) Saos-2 cells induces apoptosis in response to IFN-β treatment, which is not inhibited by the expression of dominant negative p53, suggesting that the induction of apoptosis by IFN-β requires an ARF function that is independent of p53 [200]. In addition, in a mouse xenograft model of pancreatic ductal adenocarcinoma metastasis, ARF inhibits tumor cell colonization independently of p53 [201]. These observations suggest that there are targets other than the MDM2-p53 axis for tumor suppression mediated by ARF.

#### 3.3.1. ARF Suppresses Ribosomal Biogenesis

ARF suppresses ribosomal biogenesis, thereby reducing cell proliferation (Figure 17). P19^ARF^ inhibits the production of ribosomal RNA (rRNA) and processing, independent of p53, thereby inhibiting cell proliferation [202]. The reduction of rRNA synthesis may be due, at least in part, to the binding of ARF to the rRNA gene promoter along with topoisomerase I, which functions in rRNA transcription, as shown by the chromatin immunoprecipitation assay [203]. In addition, ARF inhibits the import of the RNA polymerase I transcription termination factor (TTF-I) into the nucleolus by binding to the nucleolar localization sequence (NoLS) [204]. ARF also suppresses ribosomal biogenesis through interaction with nucleophosmin (NPM), which is thought to facilitate the maturation of pre-ribosomal particles and the nuclear export of ribosomes [205,206,207,208]. The association of ARF and NPM occurs in the absence of MDM2 and p53 and is antagonized by MDM2. The knockdown of NPM reduces the amount of ARF in the nucleolus and facilitates ARF-MDM2 interaction in the nucleoplasm, which correlates with growth suppression and p53 activation mediated by ARF. Conversely, over-expression of NPM increases nucleolar ARF and inhibits ARF function. These observations suggest that NPM targets ARF to nucleoli and inhibits its function [209]. Thus, NPM and ARF antagonize one another to either promote or inhibit cell growth. Indeed, in *Arf*(−/−) cells, ribosome biogenesis and protein synthesis are enhanced, leading to increased cell protein and volume, which are reversed by the knockdown of NPM [210]. ARF also binds to DDX5, which was originally identified as an RNA helicase and also functions as a transcriptional coactivator. ARF reduces the amount of nucleolar DDX5, which is required for transcription and maturation of rRNA, and also inhibits the association of DDX5 with the rDNA promoter and nuclear pre-ribosomes. Thus, the inhibition of DDX5 by ARF results in reduced ribosome genesis and cell proliferation [211]. Hepatocyte odd protein shuttling (HOPS), whose over-expression results in cell cycle arrest in G0/G1 and whose knockdown causes centrosome hyper-amplification, interacts with NPM and p19^ARF^, functioning as a bridging protein between NPM and p19^ARF^, which are mutually antagonistic with respect to tumor cell proliferation [212]. ARF also suppresses the expression of Drosha, a known DDX5 interacting protein, and attenuates the maturation of miRNAs and ribosomal RNAs (rRNAs), by suppressing the translation of Drosha mRNAs [213]. ARF-null cells, transformed by oncogenic Ras(V12), show an increased expression of Drosha, and the knockdown of Drosha inhibits Ras-dependent cellular transformation [214]. These observations point to the critical roles of ARF in the suppression of cell growth by inhibiting ribosome biogenesis.

#### 3.3.2. ARF Suppresses the Expression of Growth-Promoting Genes

ARF suppresses gene expression by binding to several transcription factors and coactivators to suppress both the transcription and translation of mRNAs (Figure 17).

ARF suppresses E2F, the central player in cell cycle progression [78]. ARF can inhibit E2F activity in p53(−/−) cells, indicating that the suppression of E2F activity is not dependent on p53. P14^ARF^ binds to E2F1 and suppresses its transcriptional activity [215,216,217], while p19^ARF^ binds directly to DP1, a heterodimeric partner of E2F, and inhibits the interaction between DP1 and E2F1 [218]. Since E2F1 requires DP1 for high affinity binding to growth-related target genes, this results in a reduction in E2F target gene expression and cell cycle arrest. Thus, DP1 also seems to be a critical direct target of ARF [219]. In addition, p19^ARF^ binds to E2F1–E2F3 and facilitates degradation through the proteasome, thereby reducing E2F transcriptional activity [220]. Taken together, the binding to and suppression of E2F seems to be an important function of ARF, independent of p53, for growth suppression.

ARF interacts with c-Myc independently of MDM2 or p53 and translocates c-Myc from the nucleoplasm to the nucleolus, thereby inhibiting its transcriptional activity [221]. ARF binding to DDX5 also inhibits its coactivator function. DDX5 binds to c-Myc and enhances its transcriptional activity. ARF binds to DDX5 and blocks its association with c-Myc, reducing c-Myc-mediated gene expression [222]. P14^ARF^ inhibits the transcriptional activity of hypoxia-inducible factor (HIF)-1α, which plays a crucial role in the adaptation of tumor cells to hypoxia by sequestering it into the nucleolus [223]. ARF also inhibits the transcriptional activity of nuclear factor-kappa B (NF-κB), which plays a critical role in cell survival, independent of MDM2 and p53. ARF inhibits the transcriptional activity of the NF-κB family member RelA by recruiting histone deacetylase 1 (HDAC1) [224]. ARF activates ATR and CHK1, which phosphorylate the transactivation domain of RelA at threonine 505, and it suppresses transcriptional activity [225]. ARF also binds to and suppresses protein phosphatase 1G (PPM1G), which functions as a coactivator of NF-κB [226]. In addition, p19^ARF^ suppresses vascular endothelial growth factor A (VEGFA) expression through repression of the translation of VEGFA mRNA, thereby suppressing tumor angiogenesis. Translational repression of VEGFA mRNA by p19^ARF^ depends on binding to NPM but does not require p53 [227]. P19^ARF^ binds to the anti-apoptotic transcriptional corepressor C-terminal binding protein (CtBP), leading to proteasome-dependent degradation of CtBP. Consistent with this, ARF expression and knockdown of CtBP by siRNA led to p53-independent apoptosis in colon cancer cells [228]. The pro-apoptotic genes *Bik*, *Bim*, and *Bmf* are repressed by CtBP, which is reversed by ARF [229]. P14^ARF^ binds to the androgen receptor (AR) and suppresses transactivation in prostate cancer cells [230]. P14^ARF^ interacts with the Myc-interacting zinc finger protein 1 (MIZ1) and facilitates the formation of a heterochromatic complex containing MIZ1, c-Myc, and trimethylated H3K9, leading to the repression of the genes involved in cell adhesion and signal transduction as well as the induction of apoptosis [231].

Reactive oxygen species (ROS) levels are tightly controlled by the transcription factor nuclear factor E2-related factor 2 (NRF2). In response to increased ROS levels, NRF2 activates target genes involved in the inactivation of ROS. The activation of oncogenes such as K-Ras(G12D), B-Raf(V619E) and Myc(ERT2) increases NRF2 activity to engage and enhance the antioxidant program, thereby lowering intracellular ROS [232]. ARF binds to NRF2 and inhibits the expression of its target genes, including *Slc7a11*. SLC7A11 is a key component of the cystine/glutamate antiporter, and cystine uptake is critical for glutathione synthesis to inactivate ROS. The suppression of SLC7A11 expression reduces intracellular cysteine levels and renders cells unable to cope with oxidative stress and susceptible to ferroptosis. Thus, ARF sensitizes cells to ferroptosis independent of p53 [233].

#### 3.3.3. ARF Facilitates Apoptosis at the Mitochondria

Although ARF is primarily located in the nucleolus, ARF translocates to the nucleoplasm and into the cytoplasm in response to a variety of stresses. In the cytoplasm, ARF localizes to the mitochondria (Figure 17). The mitochondrial protein p32 binds to the C-terminal region of ARF, encoded by exon 2, and recruits ARF to the mitochondria. The localization of ARF to the mitochondria reduces the mitochondrial outer membrane potential and induces apoptosis. Mutations in exon 2 of *Arf*, which are frequently observed in human cancers, compromise the localization of ARF to the mitochondria and the consequent induction of apoptosis. These observations may explain the frequent mutations of exon 2 in human cancers [234]. Interestingly, the interaction of ARF with p32 is not required for the mitochondrial accumulation of ARF, and highly hydrophobic domains within the amino-terminal half of p14^ARF^ serve as mitochondrial localization sequences. This allows ARF to interact with p32 and Bcl-xL to induce apoptosis (Figure 17) [235]. ARF also activates BAK by reducing the expression levels of MCL-1 and BCLxL [236]. The C-terminal nuclear/nucleolar localization sequence (NLS/NoLS) of p14^ARF^ is crucial for nucleolar localization of p14^ARF^ in the absence of cellular stress. Protein arginine methyltransferase 1 (PRMT1) methylates several arginine residues in the NLS/NoLS of p14^ARF^. Genotoxic stress facilitates the interaction of PRMT1 and p14^ARF^, leading to the arginine methylation of p14^ARF^. The methylation of the NLS/NoLS promotes the release of p14^ARF^ from the nucleolus, subsequently leading to the interaction with mitochondrial p32 and induction of apoptosis independent of p53 [237].

#### 3.3.4. ARF Facilitates Autophagy at the Mitochondria

Autophagy is a self-digesting process of cells, which degrades unnecessary or dysfunctional components through the lysosomal pathway, enabling energy supply and organelle renewal in response to insufficient nutrients supply [238]. Enforced autophagy often causes cell death (autophagic cell death) and contributes to tumor suppression.

ARF can induce autophagy in cells lacking p53 function [239]. Mitochondrial ARF interacts with the BCL-2 family member BCL-xL, relieving the inhibition of the Beclin-1/Vps34 complex, which is essential for autophagy (Figure 17) [240]. ARF interacts with cytosolic 70 kDa heat shock protein (HSP70), which mediates the trafficking of ARF to the mitochondria and autophagy [241]. For MDM2-mediated activation of p53, the region of ARF coded by exon 1 is crucial. In contrast, for the induction of autophagy, similar to apoptosis, ARF exon 2 is important, which contains the majority of *Arf* mutations in human cancer [242], suggesting that the induction of autophagy by ARF also plays an important role in tumor suppression. Autophagy contributes to the survival of cells by adaptation to energy starvation. Consequently, ARF can also contribute to tumor promotion according to cellular circumstances by facilitating autophagy in a p53-independent manner. It is suggested that the phosphorylation of ARF at Thr8 by protein kinase C may control whether ARF promotes or counteracts autophagy, since over-expression of a phosphomimetic mutant ARF(T8D) suppresses autophagy [243].

In addition, a short mitochondrial form of p19^ARF^ has been identified that induces autophagy and caspase-independent cell death. The short isoform (smARF, short mitochondrial ARF) is produced by translation from an internal initiation codon, Met45, and is devoid of the nucleolar functional domains. The human *p14^ARF^* mRNA also produces the shorter isoform. In response to oncogenic stress, the expression of smARF increases concomitantly with full-length ARF and reduces mitochondrial outer membrane potential without causing cytochrome *c* release or caspase activation, leading to the induction of autophagy and cell death [244]. SmARF also interacts with the mitochondrial p32 protein to stabilize the protein [245]. Full-length ARF can cause cellular autophagy, whereas smARF is suggested to selectively induce autophagy of the mitochondria (mitophagy) [242].

#### 3.3.5. ARF Contributes to Genome Stability

ARF seems to contribute to the maintenance of chromosomal stability. The nuclear interactor of ARF and MDM2 (NIAM) binds both ARF and MDM2. NIAM activates p53 in collaboration with ARF and causes a G1 phase cell cycle arrest. NIAM also inhibits cell growth in cells lacking p53, and knockdown experiments shows that NIAM is not essential for ARF-mediated growth inhibition. These results suggest that NIAM and ARF act both in the same and distinct pathways to cooperatively suppress cell growth. Intriguingly, the knockdown of NIAM accelerates chromosomal instability, indicating a role in maintaining chromosomal stability [246]. ARF also plays p53-independent roles in the mitotic checkpoint to maintain chromosomal stability. Aneuploidy is induced upon the loss of ARF function both in vitro and in vivo. ARF-null mouse embryonic fibroblasts (MEFs) show mitotic defects such as misaligned and lagging chromosomes, multipolar spindles, and tetraploidy, likely due to the over-expression of Aurora B. The mitotic defects of ARF-null cells can be restored by returning the expression of Aurora B to near-normal levels. These observations indicate that ARF plays an important role in chromosome segregation and the mitotic checkpoint by restraining Aurora B, thereby maintaining chromosomal stability [247].

#### 3.3.6. ARF Facilitates the SUMOylation of Interacting Proteins

Much of the p53-independent functions of ARF appear to be mediated by the SUMOylation of its interacting proteins. ARF induces the SUMOylation of interacting proteins, including MDM2 and NPM, independently of p53 [116]. The introduction of ARF in *Arf*(−/−) NIH 3T3 cells induces the SUMOylation of MDM2 and NPM before p53-dependent cell cycle arrest. The binding of ARF to MDM2 and NPM and the nucleolar localization of ARF is required for ARF to SUMOylate MDM2 and NPM. The inhibition of the SUMO activating enzyme (E1) or knockdown of the SUMO conjugating enzyme (E2/Ubc9) blocked ARF-induced SUMOylation of MDM2, but had no effect on ARF activation of p53. These observations suggest that the p53-independent effects of ARF on gene expression and tumor suppression might depend on the ARF-induced SUMOylation of interacting proteins [248].

P63 is a member of the p53 family, whose gene encodes two isoforms; one has a transcriptional activation domain in the N-terminal region (TAp63), and the other lacks this domain but has another transcriptional domain at the extreme N-terminus (ΔNp63). TAp63 functions in the development of tissues, such as skin, by activating target genes. In contrast, ΔNp63 suppresses not only TAp63 but also p53, thereby promoting cell growth and contributing to tumorigenesis. p14^ARF^ interacts with both TAp63 and ΔNp63 through their N-terminal trans-activation domains and inhibits p63-mediated trans-activation and trans-repression [249]. p14^ARF^ also destabilizes certain p63 isoforms. ΔNp63α is the most abundantly expressed p63 isoform, and p14^ARF^ targets ΔNp63α to proteasomal degradation by stimulating its SUMOylation. ΔNp63 is preferentially SUMOylated by SUMO2, and p14^ARF^ increases the efficiency of this process [250]. For efficient proteasomal degradation of ΔNp63α, both SUMOylation and ubiquitylation are required [251].

ARF stimulates the SUMOylation of NPM, which facilitates ribosome biogenesis, thereby suppressing ribosome biogenesis. The nucleolar SUMO-specific protease SUMO1/sentrin/SMT3 specific peptidase 3 (SENP3) associates with NPM and catalyzes the deSUMOylation of NPM-SUMO conjugates. The knockdown of SENP3 interferes with nucleolar ribosomal RNA processing and inhibits the conversion of the 32S rRNA species to the 28S form, which is also observed on the depletion of NPM. These observations suggest that SENP3 plays critical roles in ribosome biogenesis by deconjugating SUMO from NPM [252]. Dependent upon the presence of NPM, p19^ARF^ triggers sequential phosphorylation, polyubiquitination, and rapid proteasomal degradation of SENP3. Thus, p19^ARF^ and SENP3 act in opposition to each other in the SUMOylation of target proteins such as NPM [253].

Taken together, ARF fulfills a variety of MDM2-p53 independent functions in the suppression of cell proliferation (Figure 17). It is expected that the induction of *Arf* gene expression by deregulated E2F, upon oncogenic changes, not only activates p53, through the inactivation of MDM2, but also activates MDM2-p53-independent functions of ARF to suppress tumorigenesis.

### 3.4. P53-Independent Functions of MDM2 in Tumor Promotion

In addition to the inhibition of transcriptional activity and facilitating the degradation of p53, MDM2 promotes cell proliferation independent of p53. MDM2 physically interacts with pRB and inhibits its ability to suppress cell proliferation [254]. Interestingly, pRB-bound MDM2 can still bind to p53 and suppress transcriptional activation, but this does not affect transcriptional repression and promotes p53-mediated apoptosis [255]. MDM2 also binds to E2F1 and DP1 through the activation domain of E2F1. In contrast to binding to p53, MDM2 binding to E2F1/DP1 stimulates its transcriptional activity [256]. Interestingly, MDM2 suppresses apoptosis induced by E2F1/DP1 but stimulates DNA synthesis and colony formation depending on DP1 [257]. These results indicate that MDM2 not only inhibits growth suppression by p53 but also promotes cell proliferation by enhancing E2F1/DP1 activity. Consistent with this, the over-expression of MDM2 in the mammary gland of mice disturbs its normal development and morphogenesis regardless of the presence of p53. This may be due to the over-expression of MDM2 promoting the S phase in the absence of mitosis, thereby generating polyploid cells [258].

MDM2 binds to not only the p53 protein but also p53 mRNA. This interaction is through the RING domain of MDM2 and impairs its E3 ligase activity but promotes the translation of p53 mRNA [259]. ATM phosphorylates MDM2 at Ser395 and facilitates the MDM2-p53 mRNA interaction. The Ser395 phosphorylation of MDM2 also promotes its SUMOylation and nucleolar accumulation, which facilitates p53 stabilization and activation following DNA damage [260]. MDM2 can also inhibit the translation of p53 mRNA by regulating ribosomal protein L26, which binds to p53 mRNA and facilitates its translation [261]. MDM2 binds to and polyubiquitylates L26, facilitating its degradation through proteasome. MDM2 also inhibits the interaction of L26 with p53 mRNA, thereby suppressing p53 protein production. Genotoxic stress suppresses the inhibitory effects of MDM2 on L26, thereby rapidly increasing p53 protein production [262].

pRB and E2F1 bind the MRE11-RAD50-NBS1 (MRN) complex through NBS1 and facilitate DNA double-strand break repair [159]. MDM2 also binds to NBS1, independent of ARF or p53, and co-localizes with NBS1 to the sites of DNA damage. However, the over-expression of MDM2 delays DNA double-strand break repair, suggesting that MDM2 is negatively involved in the repair process and genomic stability [263].

MDM2 promotes epithelial-mesenchymal transition (EMT) and invasiveness by several mechanisms. MDM2 binds directly to E-cadherin and down regulates its expression by ubiquitination and endosomal degradation. The over-expression of MDM2 in breast cancer cells disrupts cell-to-cell contacts and increases cell motility and invasiveness [264]. MDM2 mediates the downregulation of Forkhead box O 3a (FOXO3a) expression by the RAS-extracellular signal-regulated kinase (ERK) pathway through ubiquitin-proteasome-mediated degradation of ERK-phosphorylated FOXO3a, leading to enhanced cell proliferation and tumorigenesis [265]. In addition, MDM2 enhances the expression of the transcription factor Slug, which suppresses the expression of E-cadherin and promotes EMT by binding to and stabilizing the Slug mRNA [266]. In contrast, p53 induces the expression of MDM2, which binds to Slug along with p53, and facilitates MDM2-mediated degradation of Slug, thereby suppressing cancer cell invasion [267].

MDM2 and MDM4 also bind to TAp73 and suppress its transcriptional activity; however, MDM2 and MDM4 do not facilitate the degradation of TAp73 but rather stabilize the protein [268,269,270,271]. Consistent with this, the inhibition of MDM2 or MDM4 in cells lacking p53 function causes cell cycle arrest, despite reduced levels of TAp73 [272], likely due to release from transcriptional repression. In addition, the inactivation of MDM2 in p53-deficient triple-negative breast cancer cells induces apoptosis mediated by TAp73 [273]. These observations suggest that MDM2 and MDM4 promote cell proliferation, at least in part, through the inhibition of TAp73.

Taken together, MDM2 and MDM4 have a variety of p53-independent functions, which may contribute to cell proliferation and tumorigenesis (Figure 18). It is reasonable to predict that the induction of ARF inhibits these functions of MDM2 and MDM4, in addition to the suppression of p53.

### 3.5. Non-Classical Functions of p53

#### 3.5.1. Novel p53 Targets Genes Important for Tumor Suppression

It is generally accepted that p53 plays a crucial role in tumor suppression by inducing cell cycle arrest or apoptosis, mediated by the transcriptional activation of the *p21^Cip1^* and *Puma* or *Noxa* genes, respectively. The requirement for these three genes was examined using triple knockout mice (p21−/−puma−/−noxa−/− mice) [274]. As expected, p53 could not induce cell cycle arrest, cellular senescence, or apoptosis in the cells obtained from these mice. However, these mice did not develop tumors up to 500 days, whereas p53-null mice all developed tumors by 250 days. This observation suggests that there are other important p53 targets for tumor suppression. Interestingly, DNA damage induced by γ irradiation remained longer in p53-null cells than in wild-type or triple knockout (p21−/−puma−/−noxa−/−) cells. In addition, p53-null cells could not induce expression of several p53 target genes involved in DNA repair. These observations indicate that the induction of cell cycle arrest, cellular senescence, or apoptosis is not sufficient for tumor suppression mediated by p53 in vivo and that maintaining genomic stability and likely other processes may be important [274,275].

The ShRNA-mediated knockdown of p53 target genes in vivo in tumor-prone genetic backgrounds identified several DNA repair genes. The combined knockdown of DNA repair genes (*Mlh1*, *Msh2*, *Rnf144b*, *Cav1*, and *Ddit4*) facilitated tumor formation to a similar extent as the knockdown of p53. Moreover, the knockdown of Mlh1 alone facilitated tumor formation in a wild-type background, and the over-expression of Mlh1 delayed tumor formation caused by the loss of p53. These observations indicate that DNA repair is crucial for p53-mediated tumor suppression [276] (Figure 19).

In a murine model of pancreatic ductal adenocarcinoma, the *tyrosine-protein phosphatase non-receptor type 14* (*Ptpn14*) gene was identified as a p53 target gene, which plays a crucial role in p53-mediated tumor suppression. Ptpn14 suppresses functions of the Yap oncoprotein in the Hippo pathway and is necessary for the suppression of pancreatic carcinogenesis mediated by p53 [277].

The cholesterol transport protein ATP-binding cassette, subfamily A, member 1 (ABCA1) mediates the transport of cholesterol from the plasma membrane to the endoplasmic reticulum (ER) and represses sterol regulatory element-binding protein 2 (SREBP-2), the master regulator of the mevalonate pathway. SREBP-2 transcriptionally induces the expression of hydroxymethylglutaryl (HMG)-CoA reductase, the rate limiting enzyme of that pathway. SREBP-2 is anchored in the membranes of the ER as a precursor protein and, upon a reduction in cholesterol levels, is cleaved and released as a mature form to transactivate target genes. The mevalonate pathway produces not only cholesterol but also isoprenoids, which are required for the prenylation of growth signal transmitters to localize to the cell membrane as intermediate metabolites. P53 transactivates the *ABCA1* gene and represses SREBP-2, thereby suppressing the mevalonate pathway [278] (Figure 19). The knockdown of mevalonate pathway genes, such as *HMG-CoA* reductase, and inhibition of HMG-CoA reductase by Atorvastatin suppresses murine hepatocellular carcinogenesis driven by the loss of p53. Similar to p53 loss, CRISPR/Cas9-mediated inactivation of the *ABCA1* gene in mice promotes liver tumorigenesis with enhanced SREBP-2 activation. These observations indicate that the *ABCA1* gene is an important p53 target in the suppression of liver tumorigenesis [278].

The transactivation domain 1 (TAD1) mutant of p53 (p53^25,26^) cannot activate classical p53 target genes such as *p21^Cip1^*, *Noxa*, and *Puma*, but it is able to activate a small subset of targets. Surprisingly, this mutant is fully capable of suppressing tumorigenesis in a knockin mouse model. Thus, the activation of most of the target genes is not necessary for p53 to suppress tumorigenesis [279,280]. Utilizing p53^25,26^, a novel p53-inducible gene *Zmat3* was identified by RNA interference and CRISPR/Cas9 screening. Zmat3 is an RNA-binding protein, which regulates exon inclusion in transcripts encoding proteins of various functions [281]. The most alternatively spliced mRNA by Zmat3 in colorectal cancer (CRC) cells was CD44, a cell attachment factor and stem cell marker that is involved in tumorigenesis. The inhibition of Zmat3 enhanced the inclusion of CD44 variant exons, generating oncogenic CD44 isoforms (CD44v) and enhanced CRC cell proliferation. The inhibition of p53 also increased CD44v, suggesting that the regulation of CD44 splicing by Zmat3 is critical for p53-mediated tumor suppression [282]. These observations indicate that Zmat3 is a novel RNA-splicing regulator and a crucial target gene of p53 in tumor suppression (Figure 19).

Studies utilizing hypomorphic mutants of p53, P47S, Y107H, and G334R, which cannot activate a subset of target genes, identified *Phospholipid transfer protein* (*Pltp*) as an important p53 target gene in growth suppression. The over-expression of PLTP inhibits colony formation of human cancer cell lines. Both p53 and PLTP decrease the sensitivity of cells to ferroptosis in human hepatocellular carcinoma cell line HepG2. These observations suggest that *Pltp* is a p53 target gene, which has a role in growth suppression and sensitivity to ferroptosis [283] (Figure 19).

Similarly, utilizing the variant of p53, Y107H, *Padi4* was identified as an important p53 target gene. PADI4 is an epigenetic modifier, which deiminates arginine, producing the non-natural amino acid citrulline. PADI4 enhances p53 activation of a subset of target genes, which are likely to function in immune response. Y107H mice developed spontaneous tumors and metastases. The increased expression of PADI4 reduced, while knockout accelerated tumor cell growth, which is correlated with immune responses, suggesting that PADI4 contributes to tumor suppression by enhancing immune function [284] (Figure 19).

P53 activates the *Mdm2* gene and suppresses the *Arf* gene to form a negative feedback loop. In addition, a new feedback loop has been identified, involving p53 inhibition of the expression of long noncoding RNA E2F1 mRNA stabilizing factor (EMS), which suppresses p53 expression [285]. EMS binds to cytoplasmic polyadenylation element-binding protein 2 (CPEB2) and inhibits CPEB2 association with p53 mRNA. The disassociation of CPEB2 from p53 mRNA inhibits polyadenylation and the translation of p53 mRNA. These observations reveal a novel feedback loop between p53 and EMS, which finely regulates p53 activity. They also suggest that EMS promotes tumorigenesis, at least in part, through the negative regulation of p53 via CPEB2 [285].
Figure 19Novel p53 target genes important for tumor suppression. Utilizing mouse models, novel target genes of p53, which are important for tumor suppression, have been identified. These include DNA repair gene *Mlh1*, cholesterol transport protein *Abca1*, RNA-binding protein *Zmat3*, phospholipid transfer protein *Pltp*, and epigenetic modifier *Padi4*.
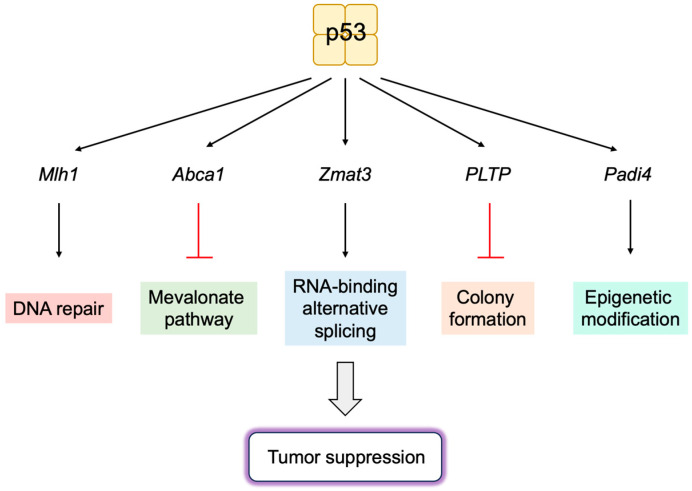


#### 3.5.2. P53 Directly Induces Apoptosis at the Mitochondria

P53 cannot induce cell cycle arrest, cellular senescence, or apoptosis in the cells obtained from triple knockout mice (p21−/−puma−/−noxa−/− mice). However, these mice are more resistant to tumorigenesis than p53-deficient mice [274]. Similarly, the p53 acetylation site mutant p53(3KR) (K117R+K161R+K162R) cannot activate most p53 target genes and cannot induce cell cycle arrest, cellular senescence, or apoptosis. However, mice with p53(3KR) are also more resistant to tumorigenesis than p53-deficient mice [286]. These results suggest that there are other functions of p53 that result in the induction of cell cycle arrest, cellular senescence, or apoptosis in vivo, independent of p53 target gene activation.

In addition to transcriptionally activating pro-apoptotic genes such as *Bax* and *Bak*, which destabilize the mitochondrial outer membrane and release cytochrome *c*, p53 directly stimulates apoptosis at the mitochondria [287]. At the outer membrane of the mitochondria, pro-survival BCL-2 family proteins BCL-xL and BCL-2 bind and inactivate pro-apoptotic BCL-2 family proteins (multidomain and BH3-only proteins), thereby stabilizing the membrane to suppress apoptosis. A portion of p53 translocates to the mitochondria and binds to BCL-xL and BCL-2 through its DNA-binding domain, inducing conformational change and releasing pro-apoptotic BCL-2 family proteins [288,289,290,291,292] (Figure 20). Two p53 molecules form a homodimer and bind one BCL-xL molecule [293]. P53 also directly activates the pro-apoptotic BCL-2 protein BAX and BAK [290,294]. P53 also binds to pro-survival BCL-2 family protein MCL-1 and disrupts the BAK-MCL-1 complex, resulting in the release and oligomerization of BAK [294] (Figure 20). This leads to the destabilization of the mitochondrial outer membrane and release of cytochrome *c* to trigger the apoptotic program.

The monoubiquitination of p53 by MDM2 facilitates translocation to the mitochondria (Figure 20). Upon arrival at the mitochondria, p53 is deubiquitinated by mitochondrial HAUSP, leading to apoptosis [295]. Hepatocyte odd protein shuttling (HOPS) binds to p53, inhibits proteasomal degradation, and facilitates the localization of p53 to the mitochondria and the induction of apoptosis [296,297]. HOPS also interferes with the nuclear transport factor importin α to increase cytoplasmic p53 levels [296].

For efficient translocation to the mitochondria and the activation of BAX, a conformational change of p53 by the prolyl isomerase Pin1 is required [298,299,300], while efficient displacement of the inhibitory MCL-1 protein from BAK requires the acetylation of p53 at Lys120 [301]. The cytoplasmic accumulation of p53 is facilitated by FOXO3a through increasing the association of p53 with the nuclear export receptor chromosomal region maintenance 1 (CRM1) [302]. Transcriptionally induced PUMA disrupts the association between p53 and BCL-xL, allowing p53 to interact with and activate BAX and BAK to promote apoptosis [303].

P53 localizes to the ER and modulates Ca^2+^ homeostasis. Upon genotoxic and oxidative stresses, p53 binds directly to the sarco/ER Ca^2+^-ATPase (SERCA) pump in the ER, reducing the oxidation of SERCA and increasing Ca^2+^ uptake into the ER. This leads to an enhanced transfer of Ca^2+^ from the ER to the mitochondria, resulting in mitochondrial Ca^2+^ overload and the induction of apoptosis [304] (Figure 20). Mitochondrial Lon is a multi-functional matrix protease, which associates with the Hsp60–mtHsp70 complex and shows chaperone activity. Upon oxidative stress, the level of Lon is increased and associates with p53 in the mitochondrial matrix, thereby inhibiting the apoptotic function of p53 [305].

Taken together, p53 directly stimulates apoptosis at the mitochondria through a variety of pleiotropic mechanisms controlled and mediated by multiple interacting factors.
Figure 20P53 directly induces apoptosis at the mitochondria. Monoubiquitinated p53 is translocated to the mitochondria, where it is deubiquitinated by HAUSP and directly induces apoptosis. P53 binds to pro-survival BCL-2 family proteins BCL-xL and BCL-2 and releases pro-apoptotic BCL-2 family proteins. Similarly, p53 binds to pro-survival BCL-2 family protein MCL-1 and disrupts the BAK-MCL-1 complex, resulting in the release and oligomerization of BAK. P53 also directly activates the pro-apoptotic BCL-2 protein BAX and BAK. P53 binds to the SERCA pump in the ER, leading to an enhanced Ca^2+^ uptake and transfer to the mitochondria, which causes apoptosis.
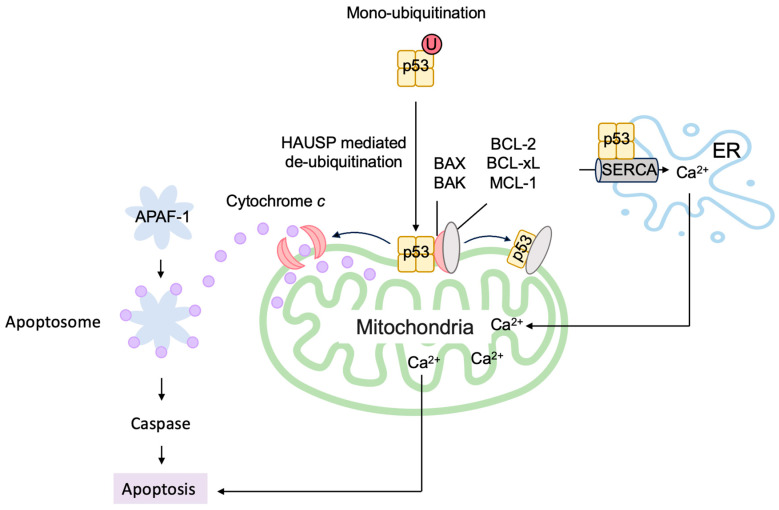


#### 3.5.3. P53 Induces Autophagy to Suppress Tumorigenesis

P53 can also induce autophagy by increasing the transcription of autophagy-related genes (*Atgs*) [300] (Figure 21). P53 induces the expression of damage-regulated autophagy modulator (DRAM), a lysosomal protein that facilitates autophagy. The knockdown of DRAM compromised p53-induced autophagy, indicating that DRAM is essential for p53-mediated apoptosis [306]. The *Dram* gene encodes a series of splice variants (SVs), of which SV4 and SV5 localize to peroxisomes and autophagosomes, respectively, and modulate autophagy [307]. DRAM3 encoded by *transmembrane protein 150B* (*Tmem150b*) also regulates both autophagy and cell death [308]. Global genomic profiling of p53-regulated genes identified a variety of autophagy-related genes, including *Atg 2*, *4*, *7*, and *10* [309]. P53 induces the expression of tumor protein 53-induced nuclear protein 1 (TP53INP1), which interacts with LC3 and ATG8-family proteins and promotes autophagy-dependent cell death [310]. P53 also induces the expression of cathepsin D, a major lysosomal aspartyl protease [311,312], which contributes to autophagy [313]. The inhibition of cathepsin D suppresses p53-dependent apoptosis and chemosensitivity [311], suggesting a role for cathepsin D in p53-dependent cell death. Upon the depletion of growth supplements, p53 transcriptionally induces the expression of transglutaminase 2, which contributes to autophagy by facilitating autophagic protein degradation and lysosomal clearance [314]. P53 causes mitochondrial defects by mitophagy and autophagic cell death by inducing the expression of DRAM and BCL-2/adenovirus E1b 19-kD protein-interacting protein 3 (BNIP-3) [315,316]. BNIP-3-induced mitophagy may also limit the metabolic shift from mitochondrial oxidative metabolism to glycolysis by maintaining mitochondrial integrity, depending on cellular circumstances [317]. Beclin-1 (ATG6) is sequestered by binding to BCL-2 and BCL-xL through its BH3 domain [318,319,320]. Beclin-1 facilitates autophagy by interacting with the autophagy-related gene products. Thus, it is expected that the induction of BH3-only proteins by p53 releases Beclin-1 from BCL-2 and BCL-xL by competing with their binding to Beclin’s BH3 domain, thereby facilitating autophagy.

Autophagy is initiated by UNC51-like autophagy-activating kinase 1 (ULK1) (Figure 21). Under glucose starvation, AMP-activated protein kinase (AMPK) activates ULK1 by phosphorylating ULK1 at Ser317 and Ser777 and facilitates autophagy. In contrast, under sufficient nutrient, mTOR inhibits ULK1 activation by phosphorylating ULK1 at Ser757 and prevents the association of AMPK with ULK1 [321]. P53 induces the expression of Sestrin1 and Sestrin2, which activate AMPK to phosphorylate TSC2, stimulating its GAP activity for GTP-bound Rheb, which is required for mTOR activity, thereby inhibiting mTOR (Figure 8) and activating ULK1 [129]. The cells with acetylation-defective mutant p53-4KR (K98R, K117R, K161R, K162R) cannot cause cell cycle arrest, senescence, apoptosis, or ferroptosis. Mice with p53-4KR are susceptible to tumor formation but resistant to early-onset tumors, which is observed in p53-null mice. However, an additional mutation at Lys136 (p53-5KR) (K98R, K117R, K136R, K161R, and K162R) compromises the resistance to early-onset tumors. Concomitant with this, p53-4KR preserves the ability to inhibit mTOR, which is lost in p53-5KR. In addition, the administration of an mTOR inhibitor inhibited the early-onset tumor formation in p53-5KR and p53-null mice. These results point to the roles of p53-mediated mTOR inhibition in tumor suppression [322].

Taken together, these observations indicate that the induction of autophagy by p53 also contributes to p53-mediated tumor suppression. However, the inhibition of p53 can also induce autophagy, which improves the survival of p53-deficient cancer cells under conditions of hypoxia and nutrient depletion, suggesting that the effects of autophagy on tumor suppression or tumor promotion may depend on cellular circumstances [323].
Figure 21P53 induces autophagy to suppress tumorigenesis. P53 induces autophagy to suppress tumorigenesis by inducing autophagy-related genes such as *Dram*, *Atg 2*, *4*, *7*, *10*, *Tp53Inp1*, *Cathepsin D*, and *Bnip-3*. P53 also facilitates the initiation of autophagy by activating UNC51-like autophagy-activating kinase 1 (ULK1) through the inhibition of mTOR.
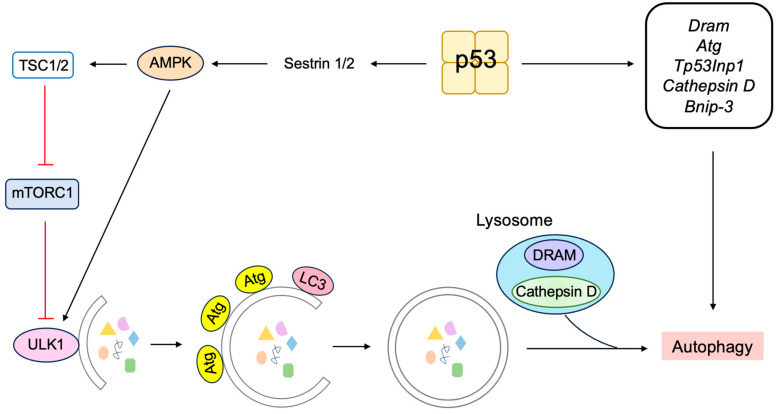


#### 3.5.4. P53 Induces Ferroptosis to Suppress Tumorigenesis

In addition to autophagic cell death, p53 regulates a unique form of cell death termed ferroptosis (iron-mediated nonapoptotic cell death), which can contribute to p53-mediated tumor suppression [324]. Ferroptosis is a type of programmed cell death, which is dependent on iron and typified by the accumulation of lipid peroxides [324,325,326,327,328,329]. The regulation of ferroptosis by p53 seems to highly depend on cellular context. P53 does not induce ferroptosis directly, and p53 modulates ferroptosis in response to iron overload or high levels of ROS. P53 can promote or suppress ferroptosis through canonical and non-canonical pathways depending on cellular context [324,328,329].

The main factor in the canonical pathway is the lipid peroxide reductase (glutathione peroxidase) GPX4 [330]. Glutathione (GSH) is a cofactor of GPX4, functioning as a reducing agent. GSH is a tripeptide composed of glutamate, cysteine, and glycine. Among them, the critical component, cysteine, is derived from cystine supplied by the cystine-glutamate antiporter system on the cell surface. The system exchanges extracellular cystine and intracellular glutamate (Figure 22).

P53 induces the expression of the mitochondrial glutaminase, glutaminase 2 (GLS2), which hydrolyzes glutamine to glutamate. Glutamate is further metabolized to α-ketoglutarate, enhancing mitochondrial respiration and ATP production. Glutamate produced by GLS2 also increases the levels GSH, since glutamate produced by GLS2 is a component of GSH, thereby decreasing ROS levels and protecting cells from oxidative stress (Figure 22) [331]. The p53P47S variant is unable to activate a subset of p53 target genes such as *glutaminase 2* (*Gls2*) and *synthesis of cytochrome c oxidase 2* (*Sco2*) involved in metabolism. In spite of an impaired ability to induce GLS2 expression, cells expressing the p53P47S variant are strikingly resistant to cell death in response to ferroptosis-inducing reagents. In addition, mice expressing p53P47S are susceptible to spontaneous tumors [332]. Moreover, cells expressing the p53P47S variant show elevated GSH levels [333], despite the inability to induce GLS2 expression. These observations suggest that other defects in metabolic regulation, such as GLS2 inhibition of PI3K/AKT signaling, may be responsible for the resistance to ferroptosis and the tumor-prone phenotype of p53P47S mice [334].

P53 also modulates ferroptosis by regulating iron metabolism through the induction of ferredoxin reductase (FDXR) expression and interaction with solute carrier family 25 member 28 (SLC25A28) (Figure 22). FDXR is a mitochondrial enzyme that transfers electrons from NADPH to ferredoxin 1 (FDX1) and FDX2 and then to cytochrome P450. FDXR also plays an important role in mitochondrial iron homeostasis by regulating the amount of iron regulatory protein 1 (IRP1) and IRP2, which bind to transferrin and ferritin mRNAs under iron-deficient conditions to facilitate or suppress its translation, respectively. Intriguingly, IRP2 also binds to p53 mRNA and suppresses its translation, thereby forming the FDXR-IRP2-p53 loop to maintain iron homeostasis [335]. In addition, p53 translocated to the mitochondria associates with SLC25A28 and increases its activity, leading to the abnormal accumulation of redox-active iron and consequent ferroptosis (Figure 22) [336].

P53 suppresses the expression of SLC7A11, which plays a key role in the cystine/glutamate antiporter system, leading to reduced cystine uptake, thereby sensitizing cells to ferroptosis (Figure 22). The acetylation-defective mutant p53(3KR) maintains the ability to repress SLC7A11 expression and to induce ferroptosis upon oxidative stress, which may explain, at least in part, the tumor suppressive phenotype observed in p53(3KR) mice. Consistent with this, SLC7A11 is often over-expressed in human tumors, and its over-expression suppresses ROS-induced ferroptosis and compromises p53(3KR)-mediated tumor suppression in xenograft models [337]. In addition, the additional mutation of an acetylation site at Lys98 in mouse p53 (Lys101 in human p53) completely abolishes the ability to repress the expression of SLC7A11, resulting in impaired ferroptosis and tumor suppression in mouse xenograft models [338]. P53 represses *Slc7a11* gene expression by an epigenetic mechanism. P53 decreases the mono-ubiquitination of histone H2B on Lys120, which facilitates transcriptional activation on the regulatory region of the *Slc7a11* gene by promoting nuclear translocation of the deubiquitinase USP7 [339]. These observations indicate a role of SLC7A11 in p53-induced ferroptosis.

P53 induces the expression of ELAV-like RNA-binding protein 1 (ELAVL1), which stabilizes a nuclear long non-coding RNA (lncRNA) LINC00336. LINC00336 functions as a sponge for miR-6852 to increase the expression of cystathionine-β-synthase (CBS) and inhibits ferroptosis [340] (Figure 22). CBS is involved in the generation of cysteine, a component of glutathione (glutamic acid, cysteine, and glycine) from homocysteine.

In the non-canonical pathway, p53 regulates ferroptosis independent of GSX4. P53 induces the expression of spermidine/spermine N1-acetyltransferase 1 (SAT1) to promote ferroptosis. SAT1 expression increases the level of arachidonate 15-lipoxygenase (ALOX15), which facilitates lipid peroxidation and sensitizes cells to ferroptosis upon oxidative stress [341] (Figure 22). In support of this, the deletion of *Alox15* in addition to *Tp53* increases tumorigenesis more than that of *Tp53* alone [342].

Under normal cellular circumstances, SLC7A11 binds to and sequesters ALOX12 and ALOXE3 from its substrate polyunsaturated fatty acids (PUFAs) [328]. Upon oxidative stress, p53 suppresses SLC7A11 expression, thereby releasing ALOX12 and ALOXE3 from SLC7A11 to oxidize PUFAs to initiate ferroptosis [343,344] (Figure 22).

According to cellular circumstances, p53 can suppress ferroptosis. The calcium-independent phospholipase iPLA2β cleaves acyl tails and releases oxidized fatty acids from phospholipids. Upon ROS-induced stress, p53 induces the expression of calcium-independent phospholipase A2 (iPLA2)β, thereby mediating the detoxification of lipid peroxides to suppress ferroptosis [345]. Upon cystine deprivation, p53 induces the expression of the CDK inhibitor p21^Cip1^, which inhibits cell cycle progression. This slows down the consumption of glutathione and delays the onset of ferroptosis [346]. These observations indicate that the effects of p53 on ferroptosis may be cellular-context-dependent.

#### 3.5.5. P53 Regulates Metabolism to Suppress Tumorigenesis

P53 also plays an important role in metabolic homeostasis [347,348,349]. In contrast to normal cells, cancer cells, in which the p53 pathway is disabled in most cases, rely on aerobic glycolysis and lactate production, even in the presence of oxygen and functional mitochondria, as an energy source for rapid cell growth (Warburg effect). P53 induces the expression of genes involved in mitochondrial respiration and suppresses the expression of genes involved in glycolysis. Thus, the loss of p53 function shifts the metabolic state from oxidative phosphorylation to the glycolytic pathway. The Warburg effect may be due, at least in part, to the loss of p53 function.

P53 induces the expression of synthesis cytochrome *c* oxidase 2 (Sco2), which is critical for the activity of the cytochrome *c* oxidase (COX) complex in the inner mitochondrial membrane, where most oxygen is utilized in eukaryotic cells. The disruption of the *Sco2* gene shifts the metabolic state toward glycolysis similar to the loss of p53 [350], indicating that the *Sco2* gene is important for p53 regulation of metabolism (Figure 23).

P53 induces the expression of TP53-induced glycolysis and apoptosis regulator (TIGAR), which lowers the levels of fructose-2,6-bisphosphate, leading to the inhibition of glycolysis [351] (Figure 23). P53 also induces the expression of Parkin, which was originally identified as a gene associated with Parkinson’s disease and is suggested to be a tumor suppressor. Although the exact mechanism has yet to be elucidated, Parkin suppresses glycolysis and facilitates mitochondrial respiration [352] (Figure 23). P53 binds to and inhibits glucose-6-phosphate dehydrogenase (G6PD), the rate-limiting enzyme of the pentose phosphate pathway (PPP) (Figure 23). Thus, the loss of p53 function enhances PPP and may increase glucose consumption for the biosynthesis of the macromolecules required for cell proliferation [353]. P53 suppresses the expression of the glucose transporters GLUT1, GLUT4, and GLUT12 [354,355] (Figure 23). In support of this, in cancer cells, the expression of several GLUTs are enhanced [356]. In addition, p53 suppresses transcription factors c-Myc and HIF-1α, which activate several glycolytic genes, thereby indirectly suppressing the glycolytic pathway [355] (Figure 23). P53 suppresses the expression of pyruvate dehydrogenase kinase-2 (PDK2) and promotes the conversion of pyruvate into acetyl-CoA instead of lactate [357]. In support of these observations, tumor cells with p53P47S, which are incapable of activating metabolic genes such as *Gls2* and *Sco2*, exhibit a reduced mitochondrial metabolism and enhanced glycolysis. P53P47S tumor cells also exhibit enhanced sensitivity to 2-deoxy-glucose, which inhibits glycolytic enzymes such as phosphoglucose isomerase and hexokinase [358].

P53 also regulates lipid metabolism. Cancer cells require enhanced synthesis or uptake of lipids for rapid proliferation [359]. P53 functions to restrain lipid metabolism. The mevalonate pathway generates cholesterol and nonsterol isoprenoids, which are required for localization of growth signal transducers to the plasma membrane. The master regulator of this pathway is the transcription factor sterol regulatory element-binding protein 2 (SREBP-2), which induces the expression of hydroxymethylglutaryl (HMG)-CoA reductase, the rate-limiting enzyme in cholesterol biosynthesis. P53 activates the *ATP binding cassette subfamily A member 1* (*Abca1*) gene (Figure 23). ABCA1 functions as a cholesterol transporter and suppresses the activation of SREBP-2, thereby decreasing flux through the mevalonate pathway [278]. Thus, p53 inhibits cholesterol synthesis and negatively affects growth signal transduction pathways. P53 also suppresses cholesterol metabolism by repressing ubiquitin-specific peptidase 19 (USP19), which deubiquitinates and stabilizes sterol O-acyltransferase (SOAT) 1 required for cholesterol esterification [360].

P53 plays a role in restraining intracellular reactive oxygen species (ROS) levels. p53 induces the expression of TIGAR, which not only represses glycolysis but also reduces intracellular ROS levels and protects cells from oxidative stress [351]. In addition, p53 induces the expression of GLS2, which catalyzes the conversion of glutamine to glutamate, a precursor of glutathione, which lowers intracellular ROS levels [361].
Figure 23P53 regulates metabolism to suppress tumorigenesis. P53 facilitates respiratory metabolism and suppresses glycolysis. P53 facilitates respiratory metabolism by the induction of Sco2, which is critical for activity of the cytochrome *c* oxidase (COX) complex, and by the suppression of the expression of PDK2. P53 suppresses glycolysis by the induction of TIGAR and Parkin and the suppression of the expression of glucose transporters GLUT1, GLUT4, and GLUT12. P53 binds to and inhibits G6PD. P53 indirectly suppresses the expression of glycolytic genes through the suppression of transcription factors c-Myc and HIF-1α, which activate the genes. p53 also suppresses cholesterol synthesis by suppressing the mevalonate pathway through the induction of ABCA1, which facilitates cholesterol transport and suppresses the activation of SREBP-2, the master regulator of HMG-CoA reductase, the rate-limiting enzyme of the pathway.
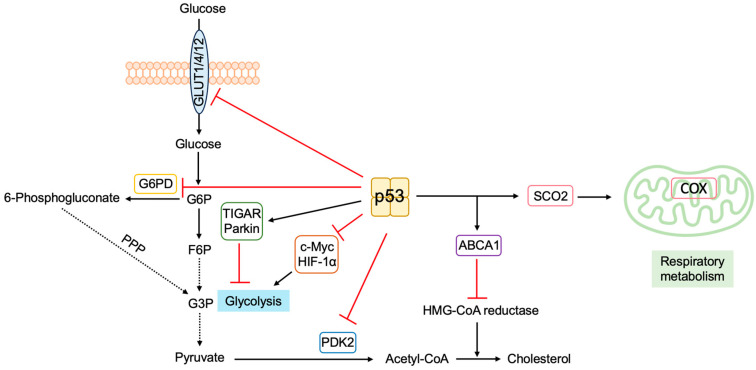


#### 3.5.6. P53 Inhibits Stemness and Promotes Differentiation

P53 is also implicated in cellular processes including self-renewal, differentiation, and reprogramming [362]. Although the precise mechanism is not yet elucidated, accumulating evidence indicates that p53 loss leads to stem-like phenotypes in cancer [363].

P53 loss facilitates the generation of induced pluripotent stem (iPS) cells [364], suggesting that p53 functions to suppress stemness. P53 activates the expression of miR-34a and miR-145, which suppress the expression of the stem cell factors octamer-binding transcription factor 4 (OCT4), Kruppel-like factor 4 (KLF4), LIN28A, and SRY-box transcription factor 2 (SOX2) [365] (Figure 24). P53-induced miR-15a and miR-16-1 downregulate the transcription factor AP4, thereby inducing mesenchymal-to-epithelial transition (MET) [366] (Figure 24). P53 suppresses self-renewal of pulmonary neuroendocrine cells, which function as stem cells, following lung injury [367]. Furthermore, p53 increases the levels of α-ketoglutarate, which results in increased levels of chromatin modification 5-hydroxymethylcytosine (5hmC) accompanied by tumor cell differentiation [368], suggesting the involvement of epigenetic gene regulation in p53-mediated differentiation (Figure 24).

Taken together, p53 has a variety of functions to suppress tumorigenesis, in addition to transcriptionally inducing cell cycle arrest and apoptosis, such as the direct induction of apoptosis at the mitochondria, induction of autophagy or ferroptosis, inhibition of cell growth by suppressing metabolism, inhibition of stemness, and facilitation of differentiation (Figure 24). It is reasonably expected that p53 activated by oncogenic changes may also fulfill at least some of these functions.
Figure 24P53 inhibits stemness and promotes differentiation. P53 inhibits stemness by the suppression of stem cell factors OCT4, KLF4, LIN28A, and SOX2 through the induction of miR-34a and miR-145. P53 increases the levels of α-ketoglutarate, resulting in increased levels of chromatin modification 5-hydroxymethylcytosine (5hmC) accompanied by tumor-cell differentiation.
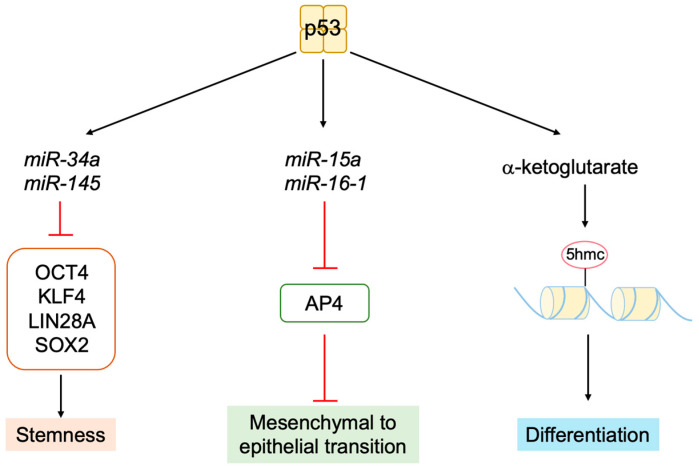


## 4. Future Directions

E2F, when activated by the loss of pRB function, induces *Arf* gene expression. It is noteworthy that E2F induced by growth stimulation does not activate the *Arf* gene, thereby maintaining normal cell growth responses. The selective induction of the *Arf* gene, by deregulated E2F, cannot be explained by the amount of free E2F (the threshold model), as shown by the knockdown of the heterodimeric partner DP. Thus, why normal growth stimulation does not induce E2F activity targeting the *Arf* gene, and how the dysfunction of RB generates deregulated E2F activity to increase ARF expression, are important questions that merit further investigation. This would facilitate our understanding of key regulatory mechanisms underlying normal cell growth and tumor suppression.

Deregulated E2F activity, leading to the activation of the *Arf* gene, is distinct from that which activates growth-related genes in that it does not depend on the heterodimeric partner DP, which is strictly required for the activation of growth-related target genes. Moreover, the sequence of the E2F-responsive element of the *Arf* gene (GC-repeats) is distinct from that of typical E2F-responsive elements of growth-related genes (TTT^C^/_G_^G^/_C_CGC). These observations suggest that the E2F that stimulates the *Arf* gene is biochemically distinct from the E2F that activates growth-related genes. The unique biochemical nature of deregulated E2F also remains to be elucidated. Deregulated E2F that activates the *Arf* tumor suppressor gene specifically exists in cancer cells due to concomitant inactivation of the p53 pathway, which tolerates deregulated E2F activity. Thus, the elucidation of the molecular nature of deregulated E2F may provide a cancer-cell-specific therapeutic target.

Deregulated E2F also activates other tumor suppressor genes, such as *Tap73* and *Bim*, which do not depend on p53 to induce apoptosis. Thus, the augmentation of deregulated E2F activity in cancer cells may sensitize them to apoptosis. Therefore, understanding the mechanism underlying deregulated E2F activity is also a critical element in the development of strategies to specifically target cancer cells. Since deregulated E2F activity is a unique characteristic of cancer cells, elucidation of these issues will facilitate the identification and implementation of future cancer-specific therapies.

The ARF-MDM2-p53 pathway, when activated, induces cellular senescence or apoptosis depending on cellular circumstances. It is not known what state of cellular circumstances determine whether cellular senescence or apoptosis will be induced. In addition, it is now evident that each component of the pathway has additional functions by interacting with other factors than the classical downstream effector and that there is an intricate interplay between these interacting factors. At present, it is difficult to predict what kind of output will be generated according to cellular circumstances when a component of the pathway such as ARF is turned on. Since the ARF-MDM2-p53 pathway is critical in tumor suppression, it is necessary to elucidate the regulatory mechanism of the intricate network of the pathway to utilize the knowledge for the treatment and prevention of cancer. For this purpose, bioinformatic analyses such as those utilized to elucidate crosstalk between metabolite production and signaling activity would be useful [369].

## 5. Conclusions

It has been well established that, upon dysfunction of the RB pathway, E2F activates the *Arf* tumor suppressor gene, thereby activating p53 to induce cell cycle arrest or apoptosis, through transcription of target genes, to protect cells from tumorigenesis (classical functions). However, accumulating evidence indicates that ARF and p53 have a variety of non-classical functions to suppress tumorigenesis (Figure 25). For example, ARF mediates p53-independent functions, such as the suppression of ribosomal biogenesis, suppression of growth-promoting transcription factors and gene expression, direct facilitation of apoptosis and autophagy at the mitochondria, and maintenance of genome stability. These functions also contribute to tumor suppression in vivo, since mice deficient in both ARF and p53/MDM2 are more susceptible to tumor formation than those deficient in p53/MDM2 alone. MDM2 also exhibits p53-independent functions, such as the suppression of apoptosis, direct facilitation of DNA synthesis and cell proliferation, and promotion of EMT and invasiveness. Therefore, ARF is thought to suppress tumorigenesis by inhibiting these p53-independent functions of MDM2. P53 fulfills transcription-independent functions, such as the direct induction of apoptosis at the mitochondria, the induction of autophagy, and ferroptosis. Evidence from transcription-deficient p53 mutant mice indicate that these functions also contribute to tumor suppression in vivo. These observations indicate important roles for non-classical functions of ARF and p53 in tumor suppression. Since ARF, induced by E2F, facilitates these functions, the regulation of the *Arf* gene by the deregulation of E2F is crucial for the activation of these alternative tumor suppressor pathways (Figure 25).

## Figures and Tables

**Figure 1 biology-12-01511-f001:**
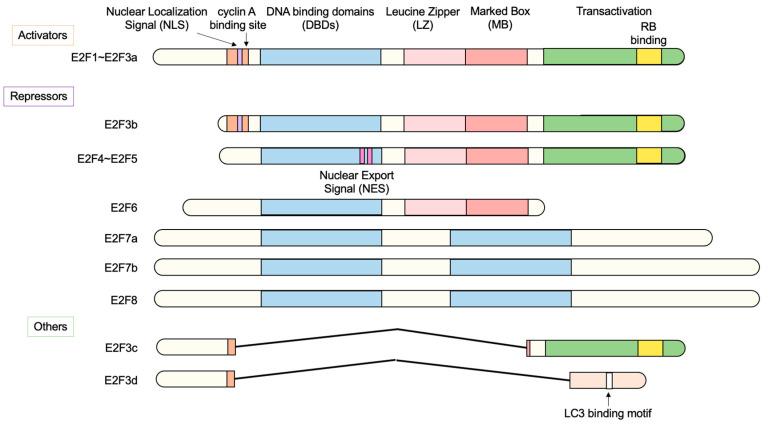
Schematic representation of E2F family members. E2F1–E2F3a are regarded as activator E2Fs, and E2F3b–E2F8 are regarded as repressor E2Fs. All family members, except the newly identified E2F3c and E3F3d, have a conserved DNA-binding domain. E2F1–E2F3 have nuclear localization signals (NLSs) and a cyclin A binding site. In contrast, E2F4 and E2F5 have a nuclear export signal (NES). E2F1–E2F5 have an RB-binding region in the transactivation domain. E2F7 and E2F8 possess two DBDs and do not depend on the heterodimeric partner DP for binding to their target sequence. E2F3c and E3F3d lack both NLSs and DBDs and are localized in the cytoplasm. E3F3d is localized at the outer membrane of the mitochondria and acts as a mitophagy receptor (described later).

**Figure 3 biology-12-01511-f003:**
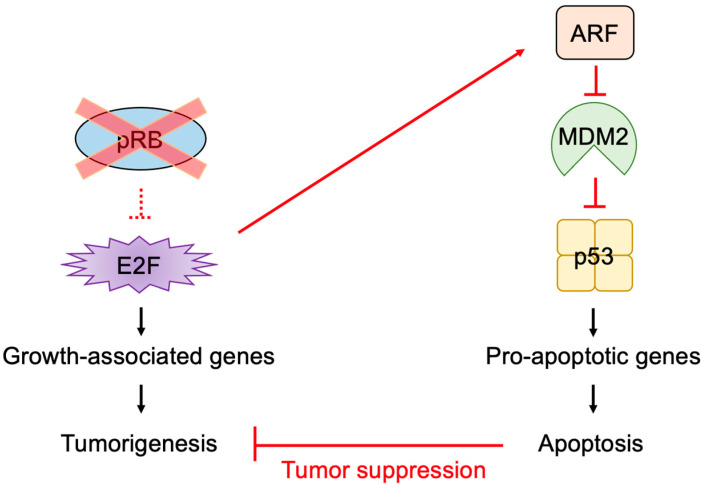
E2F links the RB pathway to the p53 pathway through the induction of ARF. Upon loss of pRB function, E2F is activated out of control by pRB and activates p53 through the induction of ARF expression.

**Figure 4 biology-12-01511-f004:**
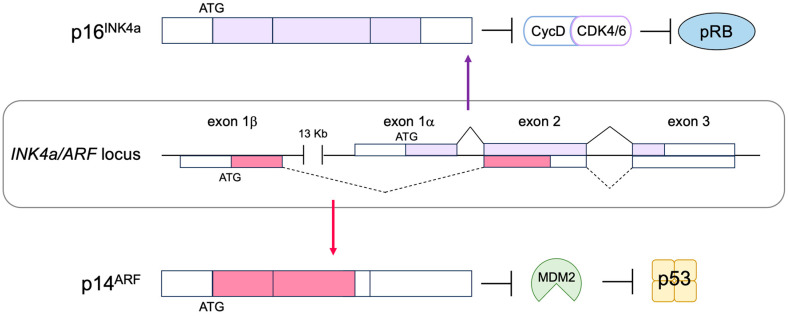
The *INK4a*/*ARF* locus. The *INK4a*/*ARF* locus encodes two products, p16^INK4a^ and p14^ARF^ (p19^ARF^ in mice), by alternative usage of promoters. p16^INK4a^ is encoded by exons 1α, 2, and 3, whereas p14^ARF^ is encoded by exons 1β, 2, and 3. p16^INK4a^ activates pRB through the inhibition of CDKs, and p14^ARF^ activates p53 through the inhibition of MDM2.

**Figure 5 biology-12-01511-f005:**
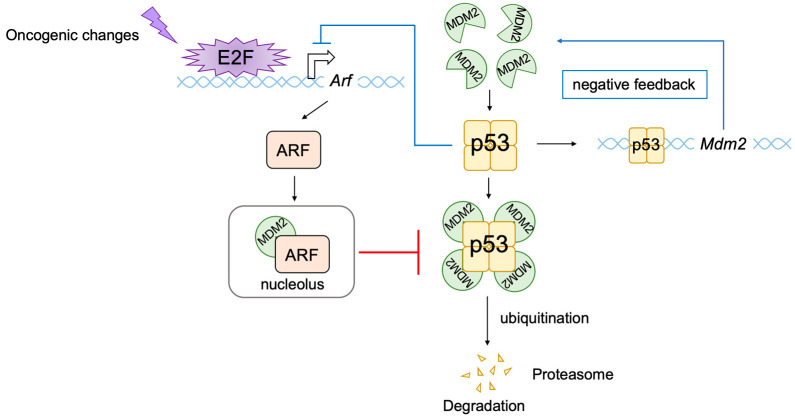
Regulatory mechanism of activation of p53 by ARF and feedback mechanism to control p53 activity. P53 is continuously ubiquitinated by MDM2 and degraded through proteasomes. ARF translocates MDM2 to the nucleolus and suppresses p53 degradation, thereby stabilizing the protein.

**Figure 6 biology-12-01511-f006:**
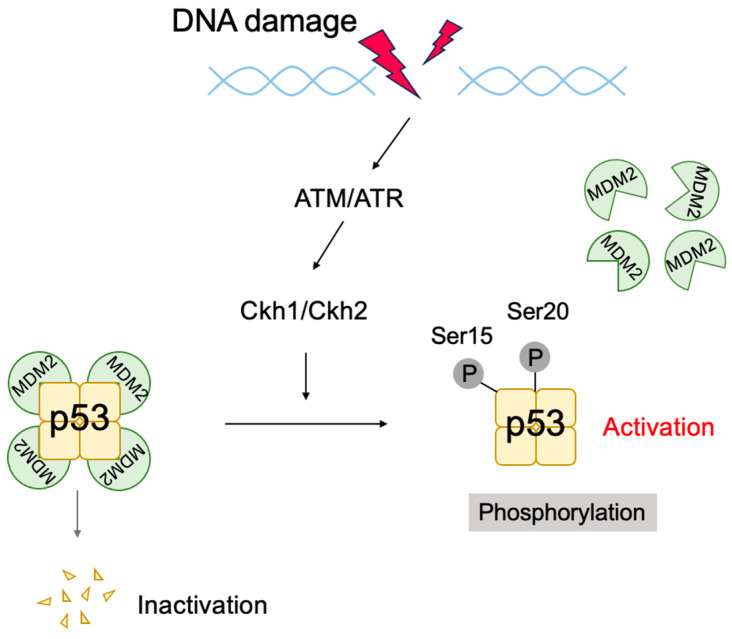
Molecular mechanism of activation of p53 by DNA damage. Upon DNA damage, ATM/ATR and CHK1/CHK2 are activated, which phosphorylate p53 at Ser15 and Ser20, thereby blocking the interaction of MDM2 and stabilizing the p53 protein.

**Figure 7 biology-12-01511-f007:**
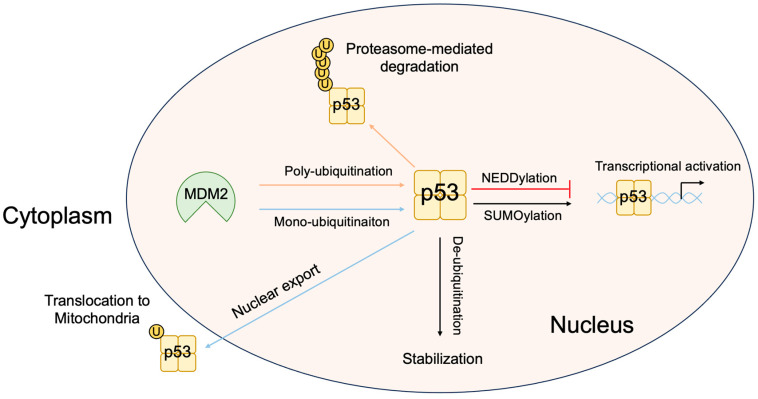
Post-translational modifications of p53. In addition to phosphorylation and ubiquitination, SUMOylation and NEDDylation also regulate p53 activity. SUMOylation enhances and NEDDylation reduces p53 activity.

**Figure 8 biology-12-01511-f008:**
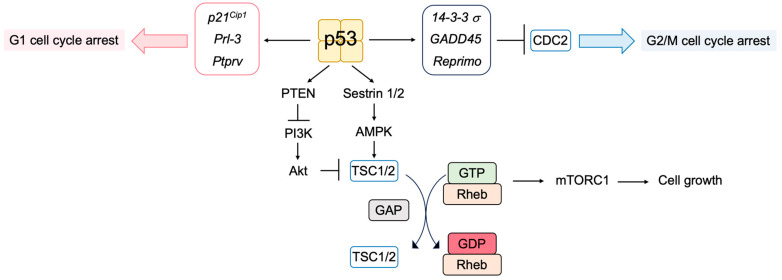
Mechanism of growth suppression mediated by p53. The P53-mediated induction of p21^Cip1^, Prl-3, and Ptprv contributes to G1 arrest, and that of 14-3-3 σ, GADD45, and Reprimo contributes to G2/M arrest. P53 also suppresses cell proliferation by inhibiting mTORC and the PI3K pathway.

**Figure 9 biology-12-01511-f009:**
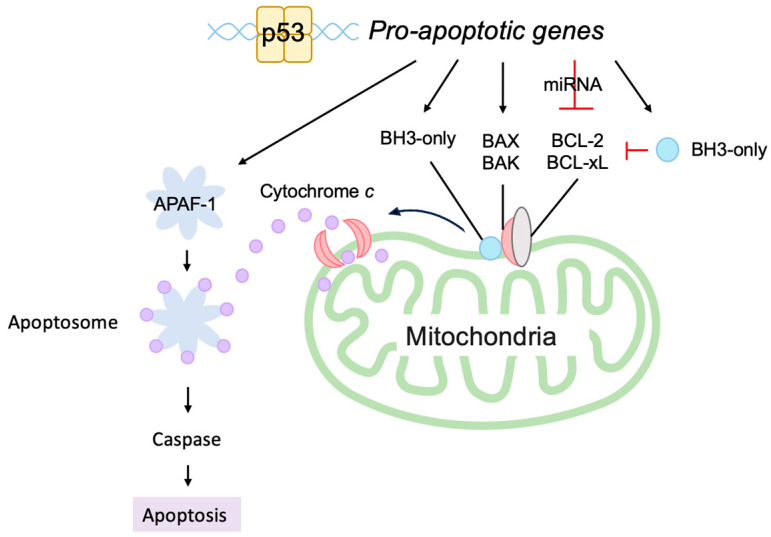
Mechanism of apoptosis induction mediated by p53. P53 induces apoptosis mainly by transcriptionally activating pro-apoptotic target genes such as *Bax*, *Noxa*, *Puma*, and *Apaf-1*. P53 also suppresses the expression of anti-apoptotic BCL-2 family members by inducing the expression of miRNAs and *Amphiregulin*.

**Figure 10 biology-12-01511-f010:**
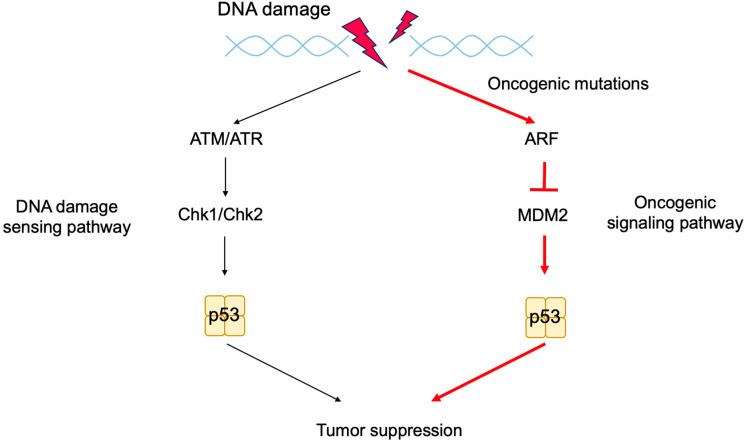
The oncogenic signaling pathway plays a crucial role in tumor suppression in vivo. The DNA-damage-sensing pathway, which causes an acute response, does not contribute to tumor suppression. In contrast, the oncogenic signaling pathway, which depends on the *Arf* tumor suppressor gene, plays a crucial role in tumor suppression.

**Figure 11 biology-12-01511-f011:**
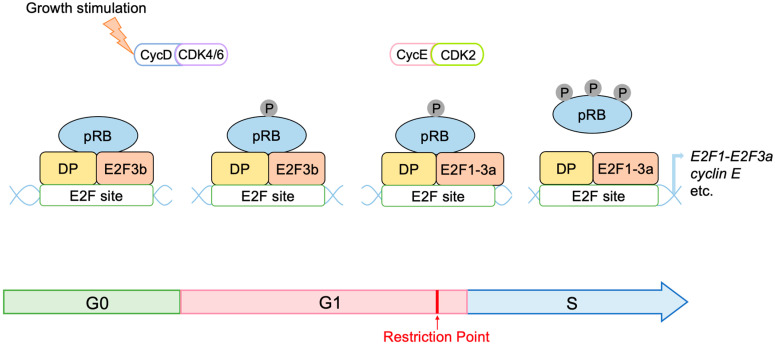
Mono-phosphorylation of pRB activates pRB to inhibit E2F. The mono-phosphorylation of pRB by cyclin D/CDK4, 6 activates pRB in binding to and suppressing E2F in the early to mid G1 phase. Additional hyper-phosphorylation by cyclin E/CDK2 inactivates pRB in the binding to and suppression of E2F at the G1/S boundary.

**Figure 14 biology-12-01511-f014:**
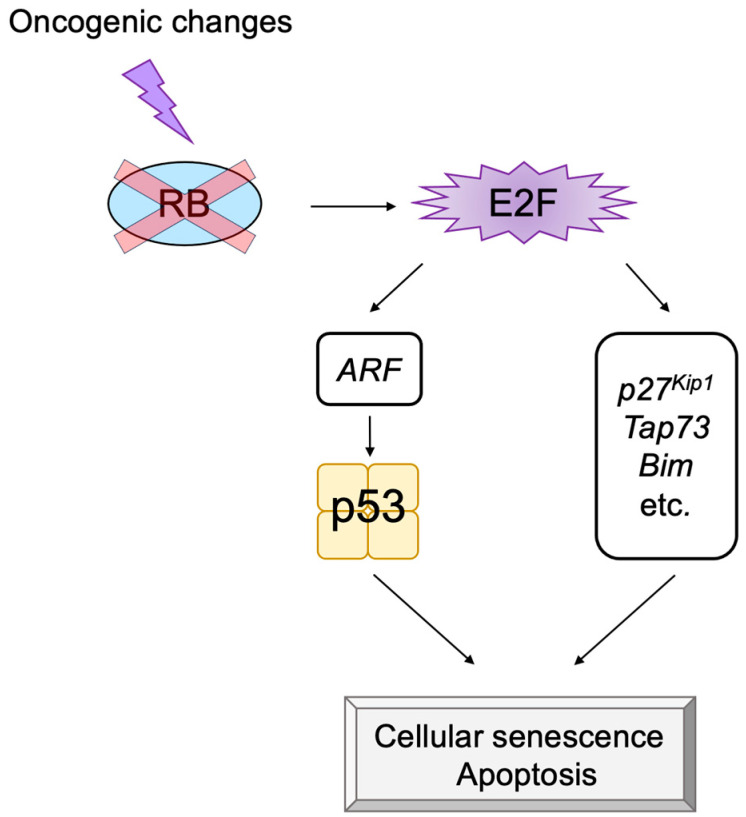
Deregulated E2F also activates p53-independent pathways. Deregulated E2F activates not only the *Arf* gene but also other tumor suppressor genes, which do not depend on p53 to induce cell cycle arrest or apoptosis. Red cross on RB indicates dysfunction of RB.

**Figure 15 biology-12-01511-f015:**
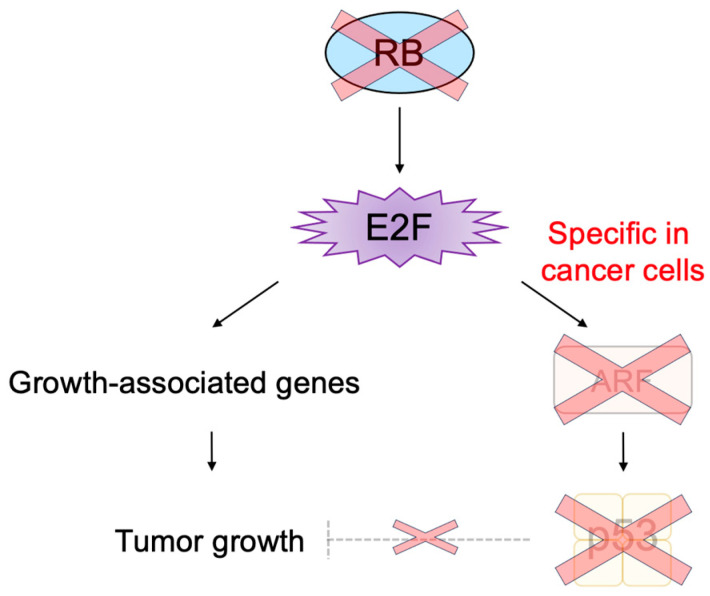
Deregulated E2F activity, which activates tumor suppressor genes, specifically exists in cancer cells. Upon dysfunction of the RB pathway, deregulated E2F1 activates the *Arf* gene to protect cells from tumorigenesis. However, the p53 pathway is also disabled in cancer cells, enabling cancer cells to survive, leaving deregulated E2F1 activity that specifically exists in cancer cells. Red crosses on RB, ARF and p53 indicate their dysfunction, and that on dashed line indicates p53 cannot suppress tumor growth.

**Figure 17 biology-12-01511-f017:**
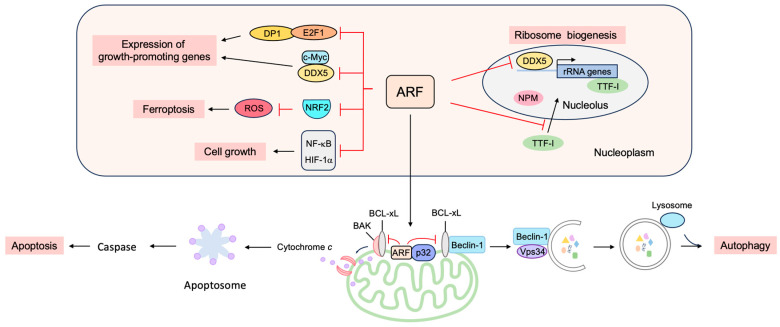
p53-independent functions of ARF in tumor suppression. ARF can suppress cell proliferation independently of p53, such as by the suppression of ribosome biogenesis and growth-promoting transcription factors and the induction of apoptosis, autophagy, and ferroptosis.

**Figure 18 biology-12-01511-f018:**
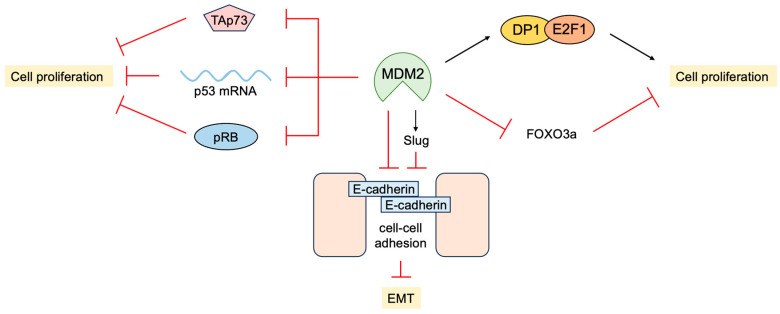
p53-independent functions of MDM2 in tumor promotion. MDM2 binds to and suppresses growth-inhibiting factors such as pRB, TAp73, and p53 mRNA. In addition, MDM2 also binds to and enhances growth-promoting factors such as E2F1/DP1. Moreover, MDM2 promotes epithelial-mesenchymal transition (EMT) by the downregulation of FOXO3a and induction of Slug expression.

**Figure 22 biology-12-01511-f022:**
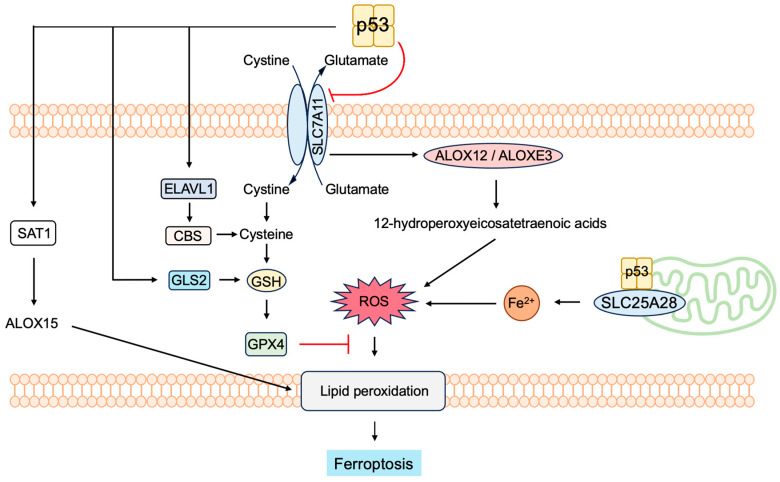
P53 can induce ferroptosis to suppress tumorigenesis. P53 can promote or suppress ferroptosis in response to iron overload or high levels of ROS depending on cellular context. P53 induces the expression of GLS2, which increases GSH and decreases ROS, thereby protecting cells from oxidative stress. P53 interacts with SLC25A28, facilitating the accumulation of iron and consequent ferroptosis. P53 suppresses the expression of SLC7A11 and inhibits cystine uptake, thereby sensitizing cells to ferroptosis. P53 increases the expression of CBS through the induction of ELAVL1 and inhibits ferroptosis. P53 induces the expression of SAT1, which induces lipid peroxidation and sensitizes cells to ferroptosis upon oxidative stress. SAT1 expression increases the level of ALOX15, which oxidizes PUFAs to promote ferroptosis. Upon ROS-induced stress, p53 suppresses SLC7A11 expression, releasing ALOX12 and ALOXE3 from SLC7A11 to oxidize PUFAs to initiate ferroptosis.

**Figure 25 biology-12-01511-f025:**
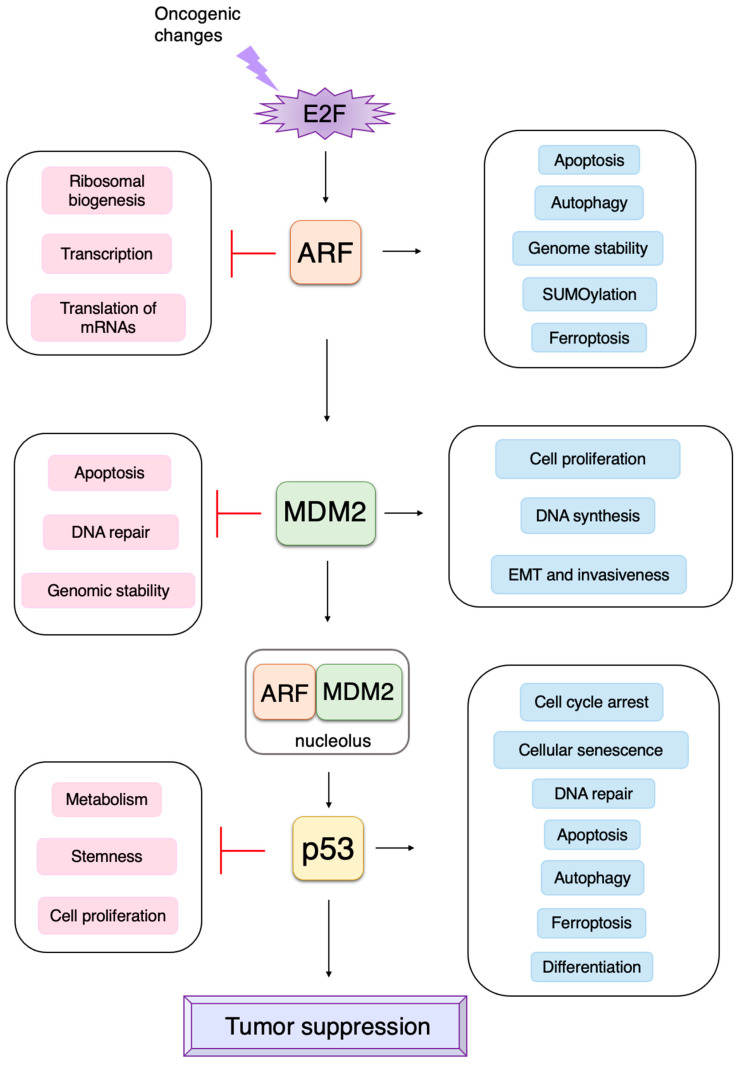
Deregulated E2F turns on multiple layers of tumor suppressive functions of the ARF-MDM2-P53 pathway. ARF, MDM2, and p53 have a variety of functions, and deregulated E2F turns on these functions by inducing the expression of ARF to suppress tumorigenesis.

## Data Availability

No new data were created or analyzed in this study. Data sharing is not applicable to this article.

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
