# Peer review of "Expanding Roles of the E2F-RB-p53 Pathway in Tumor Suppression"

_biology, 2023, doi:10.3390/biology12121511_

Round 1
Reviewer 1 Report
Comments and Suggestions for Authors
This is a comprehensive review that explains multiple layers of the E2F-RB-p53 onco-signaling cascade.
Although it is well-formatted, there is a lack of understanding of how this information can be used in translational medicine.
Therefore, I would suggest authors comment on mechanistic pathway models that can use the cellular mechanisms refined in their review: https://www.ncbi.nlm.nih.gov/pmc/articles/PMC10138666/
Such algorithms are able to integrate Omics data with the cellular networks discussed in this article, and often their accuracy depends on how well they are structured. I find this article valuable from this perspective.
Comments on the Quality of English Language
Minor editing of English language required. Authors can do it.
Author Response
According to the reviewer’s comments and the reference paper, we have added discussion to explain how this information can be used in translational medicine at the end of「4. Future Directions」with citation of the paper and highlighted it in gray.
Reviewer 2 Report
Comments and Suggestions for Authors
The authors summarized the classical and non-conventional E2F-Rb-p53 signaling pathway and its function in cancer progression. This study may be of great interest to readers in cancer. But there are several concerns listed below.
1. Please make sure all the abbreviations should be followed by their full name.
2. Figure1 the structures of E2F3c and E2F3d are not correct. No NLS domain. Make sure all the figure legends are included and in a proper style.
3. delete all the underlines in all figures.
4. line 212, protooncogenes, keep the spelling the same within the whole manuscript, proto-oncogenes
Comments on the Quality of English Language
good, but need to be improved
Author Response
- Please make sure all the abbreviations should be followed by their full name.
We have checked all the abbreviations and added their full name. They are highlighted in green.
- Figure 1, the structures of E2F3c and E2F3d are not correct. No NLS domain. Make sure all the figure legends are included and in a proper style.
We did not depict NLS domain in figure 1. The reviewer might have mistook remaining part of cyclin A binding site for NLS domain. The orange area is “cyclin A binding site” and not NLS domain. All the figure legends were included just under the title of the figure, and we have added a space between them and text to discriminate figure legends.
- delete all the underlines in all figures.
We have deleted all the underlines in all figures.
- line 212, protooncogenes, keep the spelling the same within the whole manuscript, proto-oncogenes
We have made the spelling of “proto-oncogenes” the same within the whole manuscript. They are highlighted in red.